# Can You Rely on Your Model Evaluation? Improving Model Evaluation with Synthetic Test Data

**Boris van Breugel**[*]
University of Cambridge
bv292@cam.ac.uk

**Nabeel Seedat**[*]
University of Cambridge
ns741@cam.ac.uk

**Fergus Imrie**
UCLA
imrie@g.ucla.edu

**Mihaela van der Schaar**
University of Cambridge
mv472@cam.ac.uk

## Abstract

Evaluating the performance of machine learning models on diverse and underrepresented subgroups is essential for ensuring fairness and reliability in real-world applications. However, accurately assessing model performance becomes challenging due to two main issues: (1) a scarcity of test data, especially for small subgroups, and (2) possible distributional shifts in the model's deployment setting, which may not align with the available test data. In this work, we introduce 3S Testing, a deep generative modeling framework to facilitate model evaluation by generating synthetic test sets for small subgroups and simulating distributional shifts. Our experiments demonstrate that 3S Testing outperforms traditional baselines—including real test data alone—in estimating model performance on minority subgroups and under plausible distributional shifts. In addition, 3S offers intervals around its performance estimates, exhibiting superior coverage of the ground truth compared to existing approaches. Overall, these results raise the question of whether we need a paradigm shift away from limited real test data towards synthetic test data.

## 1 Introduction

**Motivation.** Machine learning (ML) models are increasingly deployed in high-stakes and safety-critical areas, e.g. medicine or finance—settings that demand reliable and measurable performance [1]. Failure to rigorously test systems could result in models at best failing unpredictably and at worst leading to silent failures. Regrettably, such failures of ML are all too common [2–9]. Many mature industries involve standardized processes to evaluate performance under various testing and operating conditions [10]. For instance, automobiles use wind tunnels and crash tests to assess specific components, whilst electronic component data sheets outline conditions where reliable operation is guaranteed. Unfortunately, current evaluation approaches of supervised ML models do not have the same level of detail and rigor.

The prevailing testing approach in ML is to evaluate only using average prediction performance on a held-out test set. This can hide undesirable performance differences on a more granular level, e.g. for small subgroups [2, 4, 5, 11, 12], low-density regions [13–15], and individuals [16–18]. Standard ML testing also ignores distributional shifts. In an ever-evolving world where ML models are employed across borders, failing to anticipate shifts between train and deployment data can lead to overestimated real-world performance [6–8, 19, 20].

---

[*]Equal Contribution

37th Conference on Neural Information Processing Systems (NeurIPS 2023).

*However, real test data alone does not always suffice for more detailed model evaluation.* Indeed, testing can be done on a granular level by evaluating on individual subgroups, and in theory, shifts could be tested using e.g. rejection or importance sampling of real test data. The main challenge is that *insufficient amounts of test data* cause *inaccurate* performance estimates [21]. In Sec. 3 we will further explore why this is the case, but for now let us give an example.

**Example 1** *Consider the real example of estimating model performance on the Adult dataset, looking at race and age variables—see Fig. 1 and experimental details in Appendix B. There are limited samples of the older Black subgroup, leading to significantly erroneous performance estimates compared to an oracle. Similarly, if we tried to engineer a distribution shift towards increased age using rejection sampling, the scarcity of data would yield equally imprecise estimates. Such imprecise performance estimates could mislead us into drawing false conclusions about our model's capabilities. In Sec. 5.1, we empirically demonstrate how synthetic data can rectify this shortfall.*

**Aim.** Our goal is to build a model evaluation framework with synthetic data that allows engineers, auditors, business stakeholders, or policy and compliance teams to understand better when they can rely on the predictions of their trained ML models and where they can improve the model further. We desire the following properties for our evaluation framework:
**(P1) Granular evaluation**: accurately evaluate model performance on a granular level, even for regions with few test samples.
**(P2) Distributional shifts**: accurately assess how distribution shifts affect model performance.

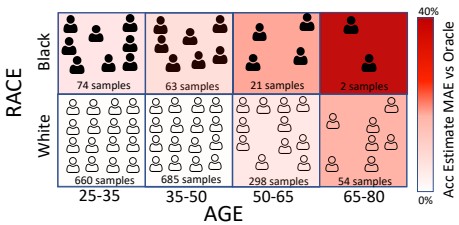

Figure 1: Limitation of testing with real data (Adult dataset): Insufficient real test samples result in imprecise performance estimates when (i) testing underrepresented subgroups (e.g. Black) and (ii) testing effect of a shift (e.g increased age)

Our primary focus is tabular data. Not only are many high-stakes applications predominately tabular, such as credit scoring and medical forecasting [22, 23], but the ubiquity of tabular data in real-world applications also presents opportunities for broad impact. To put it in perspective, nearly 79% of data scientists work with tabular data on a daily basis, dwarfing the 14% who work with modalities such as images [24].

Moreover, tabular data presents us with interpretable feature identifiers, such as ethnicity or age, instrumental in defining minority groups or shifts. This contrasts with other modalities, where raw data is not interpretable and external (tabular) metadata is needed.

**Contributions.** ① *Conceptually*, we show why real test data may not suffice for *model evaluation* on subgroups and distribution shifts and how synthetic data can help (Sec. 3). ② *Technically*, we propose the framework 3S-Testing (s.f. **S**ynthetic data for **S**ubgroup and **S**hift Testing) (Sec. 4). 3S uses conditional deep generative models to create synthetic test sets, addressing both (P1) and (P2). 3S also accounts for possible errors in the generative process itself, providing uncertainty estimates for its predictions via a deep generative ensemble (DGE) [25]. ③ *Empirically*, we show synthetic test data provides a more accurate estimate of the true model performance on small subgroups compared to baselines, including real test data (Sec. 5.1.1), with prediction intervals providing good coverage of the real value (Sec. 5.1.2). We further demonstrate how 3S can generate data with shifts which better estimate model performance on shifted data, compared to real data or baselines. 3S accommodates both minimal user input (Sec. 5.2.1) or some prior knowledge of the target domain (Sec. 5.2.2).

## 2   Related Work

This paper primarily engages with the literature on model testing and benchmarking, synthetic data, and data-centric AI—see Appendix A for an extended discussion.

**Model evaluation.** ML models are mostly evaluated on hold-out datasets, providing a measure of aggregate performance [26]. Such aggregate measures do not account for under-performance on specific subgroups [2] or assess performance under data shifts [7, 27].

The ML community has tried to remedy these issues by creating better benchmark datasets: either manual corruptions like Imagenet-C [28] or by collecting additional real data such as the Wilds

benchmark [8]. Benchmark datasets are labor-intensive to collect and evaluation is limited to specific benchmark tasks, hence this approach is not flexible for *any* dataset or task. The second approach is model behavioral testing of specified properties, e.g. see Checklist [29] or HateCheck [30]. Behavioral testing is also labor-intensive, requiring humans to create or validate the tests. In contrast to both paradigms, 3S generates synthetic test sets for varying tasks and datasets.

**Challenges of model evaluation.** 3S aims to mitigate the challenges of model evaluation with limited real test sets, particularly estimating performance for small subgroups or under distributional shifts. We are not the first to address this issue. ■ **Subgroups:** Model-based metrics (MBM [21]) model the conditional distribution of the predictive model score to enable subgroup performance estimates. ■ **Distribution shift:** Prior works aim to predict model performance in a shifted target domain using (1) Average samples above a threshold confidence (ATC [31]), (2) difference of confidences (DOC [32]), (3) Importance Re-weighting (IM [33]). A fundamental difference to 3S is that they assume access to *unlabeled data* from the target domain, which is unavailable in many settings, e.g. when studying potential effects of unseen or future shifts. We note that work on robustness to distributional shifts is not directly related, as the goal is to learn a model robust to the shift, rather than reliably estimating performance of an already-trained model under a shift.

**Synthetic data.** Improvements in deep generative models have spurred the development of synthetic data for different uses [34], including privacy (i.e. to enable data sharing, 35, 36), fairness [37, 38], and improving downstream models [39–42]. 3S provides a completely *different* and unexplored use of synthetic data: improving *testing and evaluation* of ML models. Simulated (CGI-based) and synthetic images have been used previously in computer vision (CV) applications for a variety of purposes — often to *train* more robust models [43, 44]. These CV-based methods require additional metadata like lighting, shape, or texture [45, 46], which may not be available in practice. Additionally, beyond the practical differences between modalities, the CV methods differ significantly from 3S in terms of (i) aim, (ii) approach, and (iii) amount of data—see Table 4, Appendix A.

## 3 Why Synthetic Test Data Can Improve Evaluation

### 3.1 Why Real Data Fails

**Notation.** Let $\mathcal{X}$ and $\mathcal{Y}$ be the feature and label space, respectively. The random variable $\tilde{X} = (X, Y)$ is defined on this space, with distribution $p(X, Y)$. We assume access to a trained black-box prediction model $f : \mathcal{X} \to \mathcal{Y}$ and test dataset $\mathcal{D}_{test,f} = \{x_i, y_i\}_{i=1}^{N_{test,f}} \overset{iid}{\sim} p(X, Y)$. Importantly, we do *not* assume access to the training data of the predictive models, $\mathcal{D}_{train,f}$. Lastly, let $M : \mathcal{Y} \times \mathcal{Y} \to \mathbb{R}$ be a performance metric.

**Real data does not suffice for estimating granular performance (P1).** In evaluating performance of $f$ on subgroups, we assume that a subgroup $\mathcal{S} \subset \mathcal{X}$ is given. The usual approach to assess subgroup performance is simply restricting the test set $\mathcal{D}_{test,f}$ to the subspace $\mathcal{S}$:

$$A(f; \mathcal{D}_{test,f}, \mathcal{S}) = \frac{1}{\mathcal{D}_{test,f} \cap \mathcal{S}} \sum_{(x,y) \in \mathcal{D}_{test,f} \cap \mathcal{S}} M(f(x), y). \tag{1}$$

This is an unbiased and consistent estimate of the true performance, i.e. for increasing $|\mathcal{D}_{test,f}|$ this converges to the true performance $A^*(f; p, \mathcal{S}) = \mathbb{E}[M(f(X), Y) | (X, Y) \in \mathcal{S}]$.

However, what happens when $|\mathcal{D}_{test,f} \cap \mathcal{S}|$ is small? The variance $\mathbb{V}_{\mathcal{D}_{test,f} \sim p} A(f, \mathcal{D}_{test,f}; \mathcal{S})$ will be large. In other words, the expected error of our performance estimates becomes large.

**Example 2** *To highlight how this affects Eq. 1, let us assume the very simple setting in which* $p(Y|X) = p(Y)$. *Despite Y being independent of X, Eq. 1 is not—the smaller* $\mathcal{S}$, *the higher the expected error in our estimate. In particular, for* $|\mathcal{S}| \to 0$ *(w.r.t.* $p(X)$*), almost certainly* $|\mathcal{D}_{test,f} \cap \mathcal{S}| = \emptyset$, *making Eq. 1 meaningless.*

As a result, we find that the smaller our subgroup $\mathcal{S}$, the harder it becomes to measure model $f$. At the same time, ML models have been known to perform less consistently on small subgroups [2–5], hence being able to measure performance on these groups would be most useful. Finally, by definition, minorities are more likely to form these small subgroups, and they are the most vulnerable

to historical bias and resulting ML unfairness. In other words, **model evaluation is most inaccurate on the groups that are most vulnerable and for which the model *itself* is most unreliable.**

**Real test data fails for distributional shift (P2).** If we do not take into account shifts between the test and deployment distribution, trivially the test performance will be a poor measure for real-world performance—often leading to overestimated performance [6–8]. Nonetheless, even if we do plan to consider shifts in our evaluation, for example by using importance weighting or rejection sampling based on our shift knowledge, real test data will give poor estimates. The reason is the same as before; in the regions that we oversample or overweight, there may be few data points, leading to high variance and noisy estimates. As expected, problems are most pervasive for large shifts, because these require higher reweighting or oversampling of individual points.

## 3.2 Why Generative Models Can Help

We have seen there are two problems with real test data. Firstly, the more granular a metric, the higher the noise in Eq. 1—even if the distribution is well-behaved like in Example 2. Secondly, we desire a way to emulate shifts, and simple reweighing or sampling of real data again leads to noisy estimates.

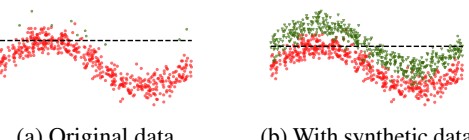

(a) Original data      (b) With synthetic data

Figure 2: **Illustration why synthetic data can give more accurate estimates using the same real samples.** Assume we want to evaluate $f$ (decision boundary=dashed line), which aims to discriminate between $Y = 1$ (green stars) and $Y = 0$ (red circles). Due to the low number of samples for $Y = 1$, evaluating $f$ using the test set alone (Eq. 1) has a high variance. On the other hand, a generative model can learn the manifold from $\mathcal{D}_{test,f}$, and generate additional data for $Y = 1$ by only learning the offset (b, green triangles). This can reduce variance of the estimated performance of $f$.

Generative models can provide a solution to both problems. As we will detail in the next section, instead of using $\mathcal{D}_{test,f}$, we use a generative model $G$ trained on $\mathcal{D}_{test,f}$ to create a large synthetic dataset $\mathcal{D}_{syn}$ for evaluating $f$. We can induce shifts in the learnt distribution, thereby solving problem 2. It also solves the first problem. A generative model aims to approximate $p(X, Y)$, which we can regard as effectively interpolating $p(Y|X)$ between real data points to generate more data within $\mathcal{S}$.

It may seem counterintuitive that $|A^* - A(f; \mathcal{D}_{syn}, \mathcal{S})|$ would ever be lower than $|A^* - A(f; \mathcal{D}_{test,f}, \mathcal{S})|$, after all $G$ is trained on $|\mathcal{D}_{test,f}|$ and there is no new information added to the system. However, even though the generative process may be imperfect, we will see that the noise of the generative process can be significantly lower than the noisy real estimates (Eq. 1). Secondly, a generative model can learn implicit data representations [39], i.e. learn relationships within the data (e.g. low-dimensional manifolds) from the entire dataset and transfer this knowledge to small $\mathcal{S}$. We give a toy example in Fig. 2. This motivates modeling the full data distribution $p(X, Y)$, not just $p(Y|X)$.

Of course, synthetic data cannot always help model evaluation, and may in fact induce noise due to an imperfect $G$. Through the inclusion of uncertainty estimates, we promote trustworthiness of results (Sec 4.1), and when we combine synthetic data with real data, we observe almost consistent benefits (see Sec. 5). In Section 6 we include limitations.

## 4 Synthetic Data for Subgroup and Shift Testing

### 4.1 Using Deep Generative Models for Synthetic Test Data

We reiterate that our goal is to generate test datasets that provide insight into model performance on a granular level (P1) and for shifted distributions (P2). We propose using synthetic data for testing purposes, which we refer to as *3S*-testing. This has the following workflow (Fig. 3): (1) train a (conditional) generative model on the real test set, (2) generate synthetic data conditionally on the subgroup or shift specification, and (3) evaluate model performance on the generated data, $A(f; \mathcal{D}_{syn}, \mathcal{S})$. This procedure is flexible w.r.t. the generative model, but a conditional generative model is most suitable since it allows precise generation conditioned on subgroup or shift information out-of-the-box. Throughout this paper we use CTGAN [47] as the generative model—see Appendix C for other generative model results and more details on the generative training process.

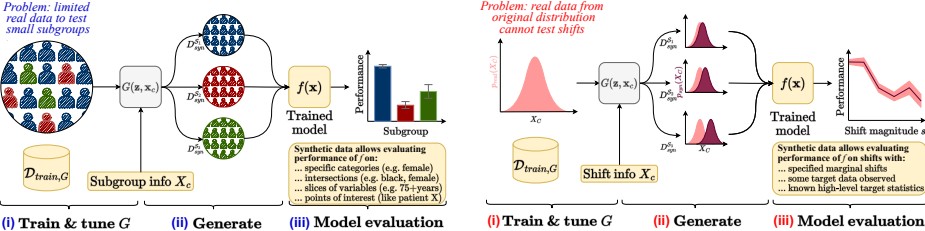

Figure 3: 3S is a framework for evaluating model performance using synthetic data generators. It has three phases: training the generative model, generating synthetic data, and model evaluation. Firstly, 3S enables (P1) *granular evaluation* when there is (i) limited real test data in small subgroups, by (ii) generating synthetic data conditional on subgroup information $X_c$, thereby (iii) permitting more reliable model evaluation even on small subgroups. Secondly, 3S enables assessment of (P2) *distributional shifts* when (i) the real test data does not reflect shifts, by (ii) generating synthetic data conditional on marginal shift information of features $X_c$, thereby (iii) quantifying model sensitivity to distributional shift. Required inputs are denoted in yellow.

**Estimating uncertainty.** Generative models are not perfect, leading to imperfect synthetic datasets and inaccurate 3S estimates. To provide insight into the trustworthiness of its estimates, we quantify the uncertainty in the 3S generation process through an ensemble of generative models [25], similar in vain to Deep Ensembles [48]. We (i) initialize and train $K$ generative models independently, (ii) generate synthetic datasets $\{\mathcal{D}_{syn}^k\}_{k=1}^K$, and (iii) evaluate model $f$ on each. The final estimates are assumed Gaussian, with statistics given by the sample mean and variance:

$$\hat{\mu}(A) = \frac{1}{K}\sum_{k=1}^{K} A(f; \mathcal{D}_{syn}^k, \mathcal{S}) \text{ and } \hat{\sigma}^2(A) = \frac{1}{K-1}\sum_{k=1}^{K}(A(f; \mathcal{D}_{syn}^k, \mathcal{S}) - \hat{\mu}(A))^2, \qquad (2)$$

which can be directly used for constructing a prediction interval. In Sec. 5.1.2, we show this provides high empirical coverage of the true value compared to alternatives.

**Defining subgroups.** The actual definition of subgroups is flexible. Examples include a specific category of one feature (e.g. female), intersectional subgroups [49] (e.g. black, female), slices from continuous variables (e.g. over 75 years old), particular points of interest (e.g. people similar to patient X), and outlier groups. In Appendix E, we elaborate on some of these further.

## 4.2 Generating Synthetic Test Sets with Shifts

Distributional shifts between training and test sets are not unusual in practice [7, 50, 51] and have been shown to degrade model performance [6, 8, 15, 52]. Unfortunately, often there may be no or insufficient data available from the shifted target domain.

**Defining shifts.** In some cases, there is prior knowledge to define shifts. For example, covariate shift [53, 54] focuses on a changing covariate distribution $p(X)$, but a constant label distribution $p(Y|X)$ conditional on the features. Label (prior probability) shift [54, 55] is defined vice versa, with fixed $p(X|Y)$ and changing $p(Y)$.[2]

Generalizing this slightly, we assume only the marginal of some variables changes, while the distribution of the other variables conditional on these variables does not. Specifically, let $c \subset \{1, ..., |\tilde{X}|\}$ denote the indices of the features or targets in $\tilde{X}$ of which the marginal distribution may shift. Equivalent to the covariate and label shift literature, we assume the distribution $p(\tilde{X}_{\bar{c}}|\tilde{X}_c)$ remains fixed ($\bar{c}$ denoting the complement of $c$).[3] Let us denote the marginal's shifted distribution by $p^s(\tilde{X}_c)$ with $s$ the shift parameterisation, with $p^0(\tilde{X}_c)$ having generated the original data. The full shifted distribution is $p(\tilde{X}_{\bar{c}}|\tilde{X}_c)p^s(\tilde{X}_c)$.

**Example: single marginal shift.** Without further knowledge, we study the simplest such shifts first: only a single $\tilde{X}_i$'s marginal is shifted. Letting $p^0(\tilde{X}_i)$ denote the original marginal, we

---

[2]Concept drifts are beyond the scope of this work.

[3]This reduces to label shift—$p(X|Y)$ constant but $p(Y)$ changed—and covariate shift—$p(Y|X)$ constant but $p(X)$ changed—for $\tilde{X}_c = Y$ and $\tilde{X}_c = X$, respectively.

define a family of shifts $p^s(\tilde{X}_i)$ with $s \in \mathbb{R}$ the shift magnitude. To illustrate, we choose a mean shift for continuous variables, $p^s(\tilde{X}_i) = p^0(\tilde{X}_i - s)$, and a logistic shift for any binary variable, $\text{logit } p^s(\tilde{X}_i = 1) = \text{logit } (p^s(\tilde{X}_i)) - s$.[4] As before, we assume $p(\tilde{X}_{\neg i}|\tilde{X}_i)$ remains constant. This can be repeated for all $i$ and multiple $s$ to characterize the sensitivity of the model performance to distributional shifts. The actual shift can be achieved using any conditional generative model, with the condition given by $\tilde{X}_i$.

**Incorporating prior knowledge on shift.** In many scenarios, we may want to make stronger assumptions about the types of shift to consider. Let us give two use cases. First, we may acquire high-level statistics of some variables in the target domain—e.g. we may know that the age in the target domain approximately follows a normal distribution $\mathcal{N}(50, 10)$. In other cases, we may actually acquire data in the target domain for some basic variables (e.g. age and gender), but not all variables. In both cases, we can explicitly use this knowledge for sampling the shifted variables $\tilde{X}_c$, and subsequently generating $\tilde{X}_{\bar{c}}|\tilde{X}_c$—e.g. sample (case 1) age from $N(50, 10)$ or (case 2) (age, gender) from the target dataset. Variables $\tilde{X}_{\bar{c}}|\tilde{X}_c$ are generated using the original generator $G$, trained on $\mathcal{D}_{test,f}$.

**Characterizing sensitivity to shifts.** This gives the following recipe for testing models under shift. Given some conditional generative model $G$, we (i) train $G$ to approximate $p(X_{\bar{c}}|X_c)$, (2) choose a shifted distribution $p^s(\tilde{X}_c)$—e.g. a marginal mean shift of the original $p^0(X_c)$ (Section 5.2.1), or drawing $x_c$ samples from a secondary dataset (Section 5.2.2); (3) draw samples $x_c$, and subsequently use $G$ to generate the rest of the variables $X_{\bar{c}}$ conditional on these drawn samples— together giving $\mathcal{D}_{syn}^s$; and (4) evaluate downstream models; (5) Repeat (2-4) for different shifts (e.g. shift magnitudes $s$) to characterize the sensitivity of the model to distributional shifts.

**More general shifts.** Evidently, marginal shifts can be generalised. We can consider a family of shifts $\mathcal{T}$ and test how a model would behave under different shifts in the family. Let $\mathcal{P}$ be the space of distributions defined on $\tilde{\mathcal{X}}$. We test models on data from $T(p)(\tilde{X})$, for all $T \in \mathcal{T}$, with $T : \mathcal{P} \to \mathcal{P}$. For example, for single marginal shifts this corresponds to $T^s(p)(\tilde{X}) = p^s(\tilde{X}_i)p(\tilde{X}_{\neg i}|X_i)$. The general recipe for testing models under general shifts then becomes as follows. Let $G$ be some generative model, we (1) Train generator $G$ on $\mathcal{D}_{train,G}$ to fit $p(X)$; (2) Define family of possible shifts $\mathcal{T}$, either with or without background knowledge; Denote shift with magnitude $s$ by $T^s$; (3) Set $s$ and generate data $\mathcal{D}_{syn}^s$ from $T^s(p)$; (4) Evaluate model on $\mathcal{D}_{syn}^s$; (5) Repeat steps 2-4 for different families of shifts and magnitudes $s$.

# 5 Use Cases of 3S Testing

We now demonstrate how 3S satisfies (**P1**) Granular evaluation and (**P2**) Distributional shifts. We re-iterate that the aim throughout is to estimate the true prediction performance of the model $f$ as closely as possible. We tune and select the generative model based on Maximum Mean Discrepancy [56], see Appendix C. We describe the experimental details, baselines, and datasets for each experiment further in Appendix B [5].

## 5.1 (P1) Granular Evaluation

### 5.1.1 Correctness of Subgroup Performance Estimates

**Goal.** This experiment assesses the value of synthetic data when evaluating model performance on minority subgroups. The challenge with small subgroups is that the conventional paradigm of using a hold-out evaluation set might result in high variance estimates due to the small sample size.

**Datasets.** We use the following five real-world medical and finance datasets: Adult [57], Covid-19 cases in Brazil [58], Support [59], Bank [60], and Drug [61]. These datasets have varying characteristics, from sample size to number of features. They also possess representational imbalance and biases, pertinent to 3S [4, 5]: ① *Minority subgroups*: we evaluate the following groups which

---

[4]We consider any categorical variable with $m$ classes using $m$ different shifts of the individual probabilities, scaling the other probabilities appropriately.

[5]Code for use cases found at: https://github.com/seedatnabeel/3S-Testing or https://github.com/vanderschaarlab/3S-Testing

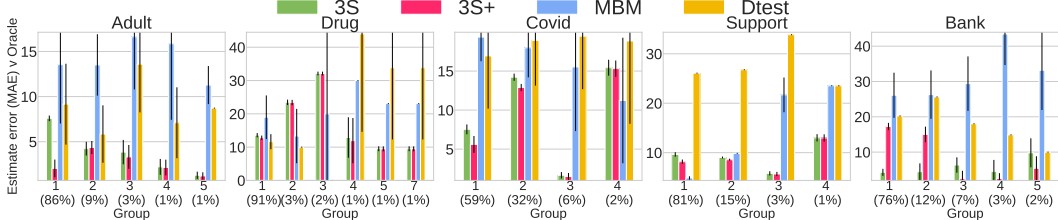

Figure 4: Assessing the reliability of performance estimates based on mean absolute error of the predicted performance estimate to the performance evaluated by oracle ($\downarrow$ better). 3S better approximates true estimates compared to both baselines. 3S+ enjoys the best of both worlds by combining synthetic and real data. We evaluate an RF model, with other model classes shown in the Appendix.

differ in proportional representation - *Adult*: Race; *Covid*: Ethnicity; *Bank*: Employment; *Support*: Race; *Drug*: Ethnicity. ② *Intersectional subgroups*: we evaluate intersectional subgroups [49] (e.g. black males or young females)— see Appendix F, intersectional model performance matrix.

**Set-up.** We evaluate the estimates of subgroup performance for trained model $f$ using different evaluation sets. We consider two baselines: (1) $\mathcal{D}_{test,f}$: a typical hold-out test dataset and (2) Model-based metrics (MBM) [21]. MBM uses a bootstrapping approach for obtaining multiple test sets. We compare the baselines to 3S testing datasets, which generate data to balance the subgroup samples: (i) *3S* ($\mathcal{D}_{syn}$): synthetic data generated by $G$, which is trained on $\mathcal{D}_{test,f}$ and (ii) *3S+* ($\mathcal{D}_{syn} \cup \mathcal{D}_{test,f}$): test data *augmented* with the synthetic dataset.

For some subgroup $\mathcal{S}$, each test set gives an estimated model performance $A(f; \mathcal{D}_., \mathcal{S})$, which we compare to a pseudo-oracle performance $A(f; \mathcal{D}_{oracle}, \mathcal{S})$: the oracle is the performance of $f$ evaluated on a large unseen real dataset $\mathcal{D}_{oracle} \sim p(X, Y)$, where $|\mathcal{D}_{oracle}| \gg |\mathcal{D}_{test,f}|$. As outlined above the subgroups are as follows: (i) Adult: Race, (ii) Drug: Ethnicity, (iii) Covid: Ethnicity (Region), (iv) Support: Race, (v) Bank: Employment status.

We evaluate the reliability of the different performance estimates based on their Mean Absolute Error (MAE) relative to the Oracle predictive accuracy estimates. We desire low MAE such that our estimates match the oracle.

**Analysis.** Fig. 4 illustrates across the 5 datasets that the 3S synthetic data (red, green) closely matches estimates on the Oracle data. i.e. lower MAE vs baselines. In particular, for small subgroups (e.g. racial minorities), 3S provides a more accurate evaluation of model performance (i.e. with estimates closer to the oracle) compared to a conventional hold-out dataset ($\mathcal{D}_{test,f}$) and MBM.

In addition, 3S estimates have reduced standard deviation. Thus, despite 3S using the same (randomly drawn test set) $\mathcal{D}_{test,f}$ to train its generator, its estimates are more robust to this randomness. The results highlight an evaluation pitfall of the standard hold-out test set paradigm: the estimate's high variance w.r.t. the drawn $\mathcal{D}_{test,f}$ could lead to potentially misleading conclusions about model performance in the wild, since an end-user only has access to a single draw of $\mathcal{D}_{test,f}$. e.g., we might incorrectly overestimate the true performance of minorities. The use of synthetic data solves this.

Next, we move beyond single-feature minority subgroups and show that synthetic data can also be used to evaluate performance on **intersectional groups** — subgroups with even smaller sample sizes due to the intersection. 3S performance estimates on 2-feature intersections are shown in Appendix F. Intersectional performance matrices provide model developers more granular insight into where they can improve their model most, as well as inform users how a model may perform on intersections of groups (especially important to evaluate sensitive intersectional subgroups).[6] Appendix F further illustrates how these intersectional performance matrices can be used as part of model reports.

We evaluate the intersectional performance estimates of 3S and baseline $\mathcal{D}_{test,f}$ using the MAE of the performance matrices w.r.t. the oracle, averaged across 3 models (i.e, RF, GBDT, MLP). The error of 3S **(11.90 $\pm$ 0.19)** is significantly lower than $\mathcal{D}_{test,f}$ **(20.29 $\pm$ 0.14)**, hence demonstrating 3S provides more reliable intersectional estimates.

---

[6]N.B. low-performance estimates by 3S only indicate poor model performance; this does not necessarily imply that the data itself is biased for these subgroups. However, it could warrant investigating potential data bias and how to improve the model.

**Takeaway.** Synthetic data provides more accurate performance estimates on small subgroups compared to evaluation on a standard test set. This is especially relevant from a representational bias and fairness perspective—allowing more accurate performance estimates on minority subgroups.

### 5.1.2 Reliability through Confidence Intervals

**Goal.** In Fig. 4, we see that all methods, including 3S, have errors in some cases, which warrants the desire to have confidence intervals at test time. 3S uses a deep generative ensemble to provide uncertainty estimates at test-time—see Sec. 4.1.

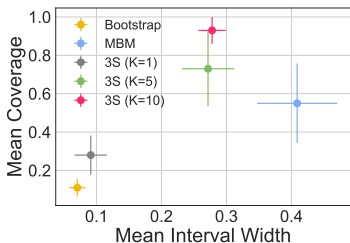

Figure 5: 3S is more reliable than baselines with higher coverage, lower widths — with less variability.

**Set-up.** We assess coverage over 20 random splits and seeds for 3S vs the baselines (B1) bootstrapping [62] confidence intervals for $\mathcal{D}_{test,f}$ and (B2) MBM: which itself uses bootstrapping for the predictive distribution. For 3S, we assess a Deep Generative Ensemble with $K = 1, 5, 10$ randomly initialized generative models. For each method, we take the average estimate $\pm 2$ standard deviations. We evaluate the intervals based on the following two metrics defined in [63–65]: (i) Coverage = $\mathbb{E}\left[1_{x_i \in [l_i, r_i]}\right]$ (ii) Width = $\mathbb{E}[|r_i - l_i|]$. Coverage measures how often the true label is in the prediction region, while width measures how specific that prediction region is. In the ideal case, we have high coverage with low width. See Appendix B for more details.

**Analysis.** Fig. 5 shows the mean test set coverage and width averaged over the five datasets. 3S (with K=5 and K=10) is more reliable, attaining higher coverage rates with lower width compared to baselines. In addition, the variability with 3S is much lower for both coverage and width. We note that this comes at a price: computational cost scales linearly with $K$. For fair comparison, we set $K = 1$ in the rest of the paper.

**Takeaway.** 3S includes uncertainty estimates at test time that cover the true value much better than baselines, allowing practitioners to decide when (not) to trust 3S performance estimates.

## 5.2 (P2) Sensitivity to Distributional Shifts

ML models deployed in the wild often encounter data distributions differing from the training set. We simulate distributional shifts to evaluate model performance under different potential post-deployment conditions. We examine two setups with varying knowledge of the potential shift.

### 5.2.1 No Prior Information

**Goal.** Assume we have no prior information for the (future) model deployment environment. In this case, we might still wish to stress test the sensitivity for different potential operating conditions, such that a practitioner understands model behavior under different conditions, which can guide as to when the model can and cannot be used. We wish to simulate distribution shifts using synthetic data and assess if it captures true performance.

**Set-up.** We consider shifts in the marginal of some feature $X_i$, keeping $p(\tilde{X}_{\neg i}|\tilde{X}_i)$ fixed (see Sec. 3). For instance, a shift in the marginal $X_i$'s mean (see Sec. 4.2). To assess performance for different degrees of shift, we compute three shift buckets around the mean of the original feature distribution: large negative shift from the mean (**-**), small negative/positive shift from the mean ($\pm$), and large positive shift from the mean (**+**).

We define each in terms of the feature quantiles. We generate uniformly distributed shifts (between min(feature) and max(feature)). Any shift that shifts the mean to less than Q1 is (**-**), any shift that shifts the mean to more than Q3 is (**+**) and any shift in between is ($\pm$).

As before, we compare estimated accuracy w.r.t. a pseudo-oracle test set. We compare two baselines: (i) Mean-shift (MS) and (ii) Rejection sampling (RS); both applied to the real data.

**Analysis.** Table 1 shows the potential utility of synthetic data to more accurately estimate performance for unknown distribution shifts compared to real data alone. This is seen both with an average lower mean error of estimates, but also across all three buckets. This implies that the synthetic data is able to closely capture the true performance across the range of feature shifts.

Table 1: Mean error in estimated accuracy across shift quantile buckets for all 5 datasets. The results show 3S indeed provides more reliable estimates even for distribution shift. ↓ is better.

| | Adult | | | | Support | | | | Bank | | | | Drug | | | | SEER | | | |
|---|---|---|---|---|---|---|---|---|---|---|---|---|---|---|---|---|---|---|---|---|
| | Mean | - | ± | + | Mean | - | ± | + | Mean | - | ± | + | Mean | - | ± | + | Mean | - | ± | + |
| **3S** | **2.6** | **2.2** | **1.8** | **3.9** | **2.0** | **2.6** | **2.0** | **1.1** | **5.4** | **3.7** | **3.6** | **6.7** | **5.6** | **5.7** | **4.4** | **7.8** | **2.7** | 5.5 | **3.0** | **2.0** |
| MS | 5.9 | 5.2 | 5.6 | 6.9 | 19.3 | 22.9 | 18.5 | 15.2 | 18.6 | 18.6 | 19.9 | 17.9 | 18.5 | 19.6 | 19.0 | 16.3 | 3.3 | **2.6** | 3.9 | 3.2 |
| RS | 15.9 | 10.5 | 17.9 | 18.6 | 25.1 | 27.8 | 24.2 | 22.3 | 18.9 | 18.9 | 20.1 | 18.3 | 20.1 | 21.7 | 21.0 | 18.2 | 20.0 | 20.3 | 23.6 | 18.6 |

**Takeaway.** Synthetic data can be used to more accurately characterize model performance across a range of possible distributional shifts.

### 5.2.2 Incorporating Prior Knowledge on Shift

**Goal.** Consider the scenario where we have *some* knowledge of the shifted distribution and wish to estimate target domain performance. Specifically, here we assume we only have access to the feature marginals in the form of high-level info from the target domain, e.g. age (mean, std) or gender (proportions). We sample from this marginal and generate the other features conditionally (Sec. 4.2). **Set-up.**

We use datasets SEER (US) [66] and CUTRACT (UK) [67], two real cancer datasets with the same features, but with shifted distributions due to com-

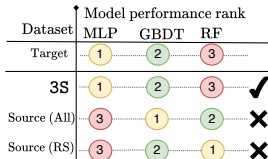

(a) Model performance rank compared.

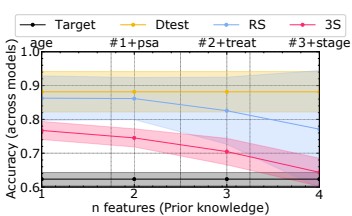

(b) Average accuracy vs increased prior knowledge

Figure 6: Incorporating prior knowledge of the shift. (a) $\mathcal{D}_{syn}$ matches the performance rank of the true target domain, which can help to select the best model to use in the target domain, and (b) $\mathcal{D}_{syn}$ better estimates target domain performance compared to baselines. Performance improves and gets closer as more prior knowledge is incorporated via added features. *Points are connected to highlight trends.*

ing from different countries. We train models $f$ and $G$ on the source domain (USA). We then wish to estimate likely model performance in the shifted target domain (UK). We assume access to information from $n$ features in the target domain (features $X_c$), sample $X_c$ from this marginal, conditionally generate $X_{\bar{c}}|X_c$. We estimate performance with $\mathcal{D}_{syn}$.

We use the CUTRACT dataset (Target) as the ground truth to validate our estimate. As baselines, we use estimates on the source test set, along with *Source Rejection Sampling (RS)*, which achieves a distributional shift through rejection sampling the source data using the observed target features. We also compare to baselines which assume access to **more information than** 3S, i.e. access to *full* unlabeled data from the target domain and hence have an advantage over 3S when predicting target domain performance. We benchmark ATC [31], DOC [32] and IM [33]. Details on all baselines are in Appendix B. Note, we repeat this experiment in Appendix E for *Covid-19 data* [58], where there is a shift between Brazil's north and south patients.

**Analysis.** In Fig. 6a, we show the model ranking of the different predictive models based on performance estimates of the different methods. Using the synthetic data from 3S, we determine the same model ranking as the true ranking on the target—showcasing how 3S can be used for model selection with distributional shifts. On the other hand, baselines provide incorrect rankings.

Table 2: 3S has lower performance estimate error in target domain for different downstream models. Rows yellow have access to more information than 3S, in the form of unlabeled data from the target domain. ↓ is better

| | mean | ada | bag | gbc | mlp | rf | knn | lr |
|---|---|---|---|---|---|---|---|---|
| 3S-Testing | **0.023** | 0.051 | **0.012** | **0.030** | **0.009** | **0.015** | **0.020** | **0.029** |
| All (Source) | 0.258 | 0.207 | 0.327 | 0.207 | 0.170 | 0.346 | 0.233 | 0.211 |
| RS (Source) | 0.180 | **0.028** | 0.298 | 0.096 | 0.014 | 0.373 | 0.213 | 0.094 |
| ATC [31] | 0.249 | 0.253 | 0.288 | 0.162 | 0.140 | 0.214 | 0.369 | 0.165 |
| IM [33] | 0.215 | 0.206 | 0.278 | 0.156 | 0.126 | 0.268 | 0.131 | 0.163 |
| DOC[32] | 0.201 | 0.207 | 0.211 | 0.162 | 0.116 | 0.223 | 0.148 | 0.161 |

Fig. 6b shows the average estimated performance of $f$ as a function of the number of observed features. We see that the 3S estimates are closer to the oracle across the board compared to baselines. Furthermore, for an increasing number of features (i.e. increasing prior knowledge), we observe that 3S estimates converge to the oracle. This is unsurprising: the more features we acquire target statistics of, the better we can model the true shifted distribution. Source RS does so too, but more slowly and with major variance issues.

We also assess raw estimate errors in Table 2. 3S clearly has lower performance estimate errors for the numerous downstream models. Beyond having reduced error compared to rejection sampling, it is interesting that 3S generally outperforms highly specialized methods (ATC, IM, DOC), which not only have access to **more information** but are also developed specifically to predict target domain performance. A likely rationale for this is that these methods rely on probabilities and hence do not translate well to the non-neural methods widely used in the tabular domain.

**Takeaway:** High-level information about potential shifts can be translated into realistic synthetic data, to better estimate target domain model performance and select the best model to use.

## 6    Discussion

**Synthetic data for model evaluation.** Accurate model evaluation is of vital importance to ML, but this is challenging when there is *limited test data*. We have shown in Sec. 3.1 that it is hard to accurately evaluate performance for small subgroups (e.g. minority race groups) and to understand how models would perform under distributional shifts using real data alone. We have investigated the potential of synthetic data for model evaluation and found that 3S can accurately evaluate the performance of a prediction model, even when the generative model is trained on the same test set. A deep generative ensemble approach can be used to quantify the uncertainty in 3S estimates, which we have shown provides reliable coverage of the true model performance. Furthermore, we explored synthetic test sets with shifts, which provide practitioners with insight into how their model may perform in other populations or future scenarios.

**Model reports.** We envision evaluations using synthetic data could be published alongside models to give insight into when a model should and should not be used—e.g. to complete model evaluation templates such as Model Cards for Model Reporting [10]. Appendix F illustrates an example model report using 3S.

**Practical considerations.** We discuss and explore limitations in detail in Appendix D. Let us highlight three practical considerations to the application of synthetic data for testing. Firstly, evaluating the performance under distributional shifts requires assumptions on the shift. These assumptions affect model evaluation and require careful consideration from the end-user. This is especially true for large shifts or scenarios where we do not have enough training data to describe the shifted distribution well enough. However, even if absolute estimates are inaccurate, we can still provide insight into trends of different scenarios. Secondly, synthetic data might have failure modes or limitations in certain settings, such as cases where there are only a handful of samples or with many samples. Thirdly, training and tuning a generative model is non-trivial. 3S' DGE mechanism mitigates this by offering an uncertainty estimate that provides insight into the error of the generative learning process. The computational cost of such a generative ensemble may be justified by the cost of untrustworthy model evaluation.

## Acknowledgements

Boris van Bruegel is supported by ONR-UK. Nabeel Seedat is supported by the Cystic Fibrosis Trust. We would like to warmly thank the reviewers for their time and useful feedback. The authors also thank Alicia Curth, Alex Chan, Zhaozhi Qian, Daniel Jarrett, Tennison Liu and Andrew Rashbass for their comments and feedback on an earlier manuscript.

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
