# Appendix: Can You Rely on Your Model? A Case for Synthetic-Data-Based Model Evaluation

## Table of Contents

# A   Extended Related Work

We present a comparison of our framework, 3S , and provide further contrast to related work. Table 3, highlights that related methods often do not permit automated test creation, in particular, they are often human labor-intensive —see Table 3. These benchmark tasks are tailored to specific datasets and tasks and hence cannot be customized and/or personalized to an end-user's specific task, dataset, or trained model. Finally, both benchmarking methods and behavioral testing require additional data to be collected or created. This contrasts 3S  which only requires the original test data, already available.

Table 3: Comparison of 3S  to human-crafted testing frameworks. (i) Automated test creation, (ii) Tests can be personalised to user dataset and use case(e.g. distributional shift), and (iii) No additional data or info required.

| Method | Approach | Assumption | Use cases | (i) | (ii) | (iii) |
|---|---|---|---|---|---|---|
| 3S  Testing (Ours) | Synthetic test data | Generative model fits the data | (P1) Subgroup testing
(P2) Distributional shift testing | ✔ | ✔ | ✔ |
| **BENCHMARK TASKS** | | | | | | |
| Imagenet-C/P [28] | Create corrupted images | Synthetic corruption reflects the real-world | Image corruption testing | ✗ | ✗ | ✗ |
| Wilds [8] | Collect data with real shifts | Collected 'wild' data reflects sufficient use cases | Distribution shift testing | ✗ | ✗ | ✗ |
| **MODEL BEHAVIOURAL TESTING** | | | | | | |
| CheckList [29] | Human crafted test scenarios | Know a-priori scenarios to test | Crafted scenario tests | ✗ | ✔ | ✗ |
| HateCheck [30] | Human crafted test scenarios | Know a-priori scenarios to test | Crafted scenario tests | ✗ | ✔ | ✗ |
| AdaTest [68] | GPT-3 creates tests, human refines | Human-in-the-loop | Weakness probing | ✔ | ✔ | ✗ |

**Data-centric AI.** The usage and assessment is vital for ML models, yet is often overlooked as operational [69, 70]. The recent focus on data-centric AI [9, 71] aims to build systematic tools to improve the quality of data used to train and evaluate ML models [64, 72]. 3S  contributes to this nascent body of work, specifically around the usage and generation of data to better evaluate and test ML models

**Simulators and counterfactual generators.** Simulators have been used to benchmark algorithms in settings such as causal effect estimation and sequential decision-making [73]. The work on simulators is not directly related, as the goal is often to test scenarios that are not available in real-world data. For example, in [73], the goal is customization of the decision-making policy in order to evaluate methods to better understand human decision-making. Similarly, there has been work on generating realistic synthetic data in the causal inference domain [74–76], but these methods focus on benchmarking causal inference methods, do not consider distributional shifts nor subgroup evaluation, and do not explore the value of the synthetic data beyond realistic, ground-truth counterfactuals.

**Synthetic data in computer vision.** Synthetic data has been used in computer vision both to improve model training and to test weaknesses in models. These methods can be grouped as follows by their motivations:

- Generate synthetic data for training, to reduce the reliance on collecting and annotating large training sets— This is different from 3S  as they focus on constructing better training sets, rather than constructing better test sets for model evaluation

- Generate synthetic data to improve the model by augmenting the real dataset with synthetic examples—*Again, this is different from 3S  as they focus on training better-performing models, rather than the evaluation of an already trained model

- Generate synthetic data to probe models on different dataset attributes. For example, in face recognition, how the model might perform on faces with long vs short hair. This is most similar to 3S , but there are clear differences in both the goal for and approach to generating the data. We compare 3S  to (i) CGI- or physics-based simulators and (ii) deep generative models for probing in computer vision.

In Table 4, we contrast both simulators and generative approaches where synthetic data to probe models on different dataset attributes.

Table 4: Comparing 3S to computer vision synthetic data approaches. $A$ is a performance metric (Eq.1) and $f$ the trained predictive model. $\mathcal{D}_{syn}(x)$ denotes synthetic data is created dependent on $x$

| Examples | Data and/or generator input | Conditioning info | Does not require pre-trained simulator/generator | $\mathcal{D}_{syn}$ used for training or testing | Goal |
|---|---|---|---|---|---|
| 3S - Subgroup | $\mathcal{D}_{train,G} = \mathcal{D}_{test,f}$ (Any dataset) | Subgroups $\mathcal{S}$ | ✓ | Testing | Reliable subgroup performance estimates for $f$, i.e. choose $\mathcal{D}_{syn} \sim p_G$ s.t. $A(f; \mathcal{D}_{syn}, \mathcal{S})] \approx \mathbb{E}_{\mathcal{D} \sim p_R}^{iid}[A(f; \mathcal{D}, \mathcal{S})]$ |
| 3S - Shift | | Shift information $T$ | | | Estimate performance of $f$ under shift $T$, i.e. choose $\mathcal{D}_{syn} \sim p_G$ s.t. $A(f; \mathcal{D}_{syn}, \mathcal{S})] \approx \mathbb{E}_{\mathcal{D} \sim p^T}^{iid}[A(f; \mathcal{D}, \mathcal{S})]$ |
| **Computer Vision** | | | | | |
| [43] | Video game engine (GTA5) Real-world data | Scene info $\mathcal{S}$ in virtual world | × | Training | Improve crowd counting performance on diff. scenes by generating semi-synthetic data for training $f$, i.e. $\max_f A(f; \mathcal{D}_{test}(\mathcal{S}))$ |
| [44] | $\mathcal{D}_{train,f}$ = VGGFace | Identity attributes | ✓ | Training | Improve overall performance of facial recognition i.e. $\max(A(f; \mathcal{D}_{test}))$ |
| [77] | 3D face model | Nuisance transforms $\mathcal{N}$ | × | Testing | Report face recognition robustness to different nuisances $\mathcal{N}$, $\mathcal{D}_{syn}(N)$ and report $A(f; \mathcal{D}_{syn}(N)), \forall N \in \mathcal{N}$ |
| [45] | 3D face model | Simulator parameters $\rho$ | × | Testing | Find adversarial failures for face recognition, i.e. find $\rho = \arg\min_\rho A(f; D_{syn}(\rho))$ |
| [46] | CityEngine, Unreal Engine, CARLA | Weather conditions $\mathcal{S}$ | × | Testing | Report segmentation performance for self-driving cars under different weather conditions, i.e. $\mathcal{D}_{syn}(S)$ and report $A(f; \mathcal{D}_{syn}(S)), \forall S \in \mathcal{S}$ |
| [78] | Pretrained StyleGAN | Implicit attributes $S$ (e.g. age, lighting) | × | Testing | Find attributes with poor performance, i.e. $\arg\min_S(A(f; \mathcal{D}_{syn}(S)))$ |
| [79] | $\mathcal{D}$ = MS-CELEB-1M | Subgroups $S$ | ✓ | Testing | Find $S$ with poor face recognition performance, i.e. $\arg\min_S(A(f; \mathcal{D}_{syn}(S))$ |

**Synthetic data and tabular approaches.**

We contrast 3S to two works DataSynthesizer [80] and AITEST [81], which while seemingly similar have specific differences to 3S . A side-by-side contrast is presented in Table 5.

**Data Synthesizer**

We believe 3S is significantly different from DataSynthesizer, in terms of aims, assumptions, and algorithmically.

***Aim and assumptions.*** Data Synthesizer primarily focuses on privacy-preserving generation of tabular synthetic data. The closest component to our work is the extension the paper proposes around adversarial fake data generation. While there are no experiments, the adversarial fake data consists of three areas. We contrast them to 3S .

The major difference is Data Synthesizer assumes access to full knowledge about the shift/distributional change. In contrast, 3S operates in a different setting - (1) No prior knowledge on the shift and (2) high-level partial knowledge about the shift through observing some variables in the target domain.

1. Edit the distribution: this assumes the user knows exactly the shift [Full knowledge of the shift]. 3S covers two different settings: (1) No prior knowledge on the shift, where only minimal assumptions on means of variables allow us to create characteristic curves like in Section 5.2. and (2) Incorporating prior knowledge, in which some features are observed from the shifted distribution and we use these to generate the full data from the shifted distribution, like in Section 5.2.2. Consequently, the difference is that 3S tackles the no and partial information settings, whereas Data Synthesizer tackles the full info setting of editing the distribution.

2. Preconfigured pathological distributions — this requires full and exact knowledge about the shift, which differs from 3S of partial knowledge and no prior knowledge settings.

3. Injecting missing data, extreme values — either such an approach is possible to incorporate in 3S . We see these ideas as complementary.

***Algorithmic.***

The authors propose three methods, one with random features, one with independent features, and one with correlated features. Due to the absence of correlation in the first two, these reduce the data utility. Let us thus focus on the third method, that does include correlation. This approach uses Bayesian Networks and is only applicable to discrete data, hence needing to discretize continuous variables. This loses utility when a coarse discretization is chosen, while a fine discretization is often intractable and data-inefficient due to the ordinal information being lost, e.g. results for $age = 31$ and $age = 32$ will generally be similar—exactly the reason why the independent approach was also introduced. Bayesian Nets are also limited in other ways, e.g. results can be influenced by the feature generation order deviating from the real data generation process' ordering, as indicated by the authors of DataSynthesizer in Figures 5 and 6.

**AITEST**

We contrast 3S to AITEST in terms of aims and assumptions, algorithm, and use cases.

***Aims and assumptions.*** AITEST has a significantly different aim and method compared to 3S . As mentioned by the reviewer, AITEST can test for adversarial robustness by generating realistic data with user-defined constraints, but this is different from our work that aims to generate synthetic test data for granular evaluation and distributional shifts.

Additionally, the assumptions on user input are quite different: AITEST enables users to define constraints on features and associations between features, whereas 3S requires information in terms of which subgroups to test or shifts to generate. We do see possibilities to combine both frameworks, e.g. through including constraints similar to the ones AITEST uses within the 3S method, or using fairness as a downstream task.

We have taken a step in this direction and added fairness as an additional experiment and have included this experiment in the new Appendix D.4.

*Algorithmic* AITEST requires a decision tree surrogate of the black-box model, whilst 3S does not need to model the black-box predictive model. AITEST defines data constraints by fitting different distributions to the features and using statistical testing to select the correct distribution. The dependencies are then captured by a DAG. 3S does not require predefined constraints and dependencies, but aims to learn these implicitly with the generative model.

*Use cases*

- Group fairness: AITEST aims to probe if a model does have a group fairness issue or not. The goal of 3S is different — even if models don't have group bias issues, with 3S we desire reliable performance metric estimates (accuracy or even fairness) which are similar to the oracle estimates on small and intersectional subgroups for which we have limited real test data.

- Adversarial robustness: AITEST does this by generating more inputs in the neighborhood of a specific sample and seeing if they behave the same. In reality, this is analogous to group-wise testing with $n = 1$, a very specific type of group testing. In contrast, with 3S we explore multiple definitions of groups from specific sensitive attributes, to intersectional groups, to points of interest (i.e. $n = 1$), to high- and low density regions.

- AITEST does not account for distribution shift, unlike 3S which looks at distribution shift with no prior knowledge and high-level knowledge.

Table 5: Comparing 3S to other tabular approaches of generating synthetic test data. $f$ is the trained predictive model and we abbreviate $A(f; \mathcal{D}) = A(f; \mathcal{D}, \Omega)$ for evaluating $f$ over all of $\mathcal{D}$ (Eq. 1). (i) used for evaluating subgroups, (ii) used for evaluating shifts, (iii) does not require discretization of continuous features, (iv) does not require modeling black-box $f$

| Examples | Inputs | (i) | (ii) | (iii) | (iv) | Generator Type | Goal |
|---|---|---|---|---|---|---|---|
| 3S | $\mathcal{D}_{train,G} = \mathcal{D}_{test,f}$ (Any dataset) Subgroups: $S$, Shifts: No/Partial knowledge | ✓ | ✓ | ✓ | ✓ | GAN | Reliable subgroup performance estimates for $f$, i.e. choose $\mathcal{D}_{syn} \sim p_G$ s.t. $A(f; \mathcal{D}_{syn}, \mathcal{S})] \approx \mathbb{E}_{\mathcal{D}^{iid}_{p_R}}[A(f; \mathcal{D}, \mathcal{S})]$ |
| | | | | | | | Estimate performance of $f$ under shift $T$, i.e. choose $\mathcal{D}_{syn} \sim p_G$ s.t. $A(f; \mathcal{D}_{syn}, \mathcal{S})] \approx \mathbb{E}_{\mathcal{D}^{iid}_{p^s}}[A(f; \mathcal{D}, \mathcal{S})]$ |
| DataSynthesizer [80] | Privacy-sensitive $\mathcal{D}$ Full knowledge of shift | × | ✓ | × | ✓ | Bayesian network | Generate private data, extensions for generating pathological data through (i) manual editing, (ii) inserting extreme values/missingness, and (iii) . |
| AITEST [81] | $\mathcal{D}_{train,G} = \mathcal{D}_{train,f}$, Constraints and dependencies (DAG) | ✓ | × | ×/✓ | × | Sample features sequentially following DAG | Subgroup performance, e.g. fairness between two sensitive groups $(S_1, S_2)$, i.e. $\frac{A(f; \mathcal{D}_{Syn}, S_1)}{A(f; \mathcal{D}_{Syn}, S_2)}$ |

# B  Experimental Details

This appendix includes details on the experiments, including (i) the datasets, and (ii) the different settings of the experiments, including the implementation of baselines.

## B.1  Datasets

Here we describe the real-world datasets used in greater detail.

**ADULT Dataset** The ADULT dataset [57] has 32,561 instances with a total of 13 attributes capturing demographic (age, gender, race), personal (marital status) and financial (income) features amongst others. The classification task predicts whether a person earns over $50K or not. We encode the features (e.g. race, sex, gender, etc.) and a summary can be found in Table 6.

Note that there is an imbalance across certain features, such as across different race groups, which is what we evaluate.

Table 6: Summary of features for the Adult Dataset [57]

| Feature | Values/Range |
|---|---|
| Age | $17 - 90$ |
| education-num | $1 - 16$ |
| marital-status | $0, 1$ |
| relationship | $0, 1, 2, 3, 4$ |
| race | $0, 1, 2, 3, 4$ |
| sex | $0, 1$ |
| capital-gain | $0, 1$ |
| capital-loss | $0, 1$ |
| hours-per-week | $1 - 99$ |
| country | $0, 1$ |
| employment-type | $0, 1, 2, 3$ |
| salary | $0, 1$ |

**Covid-19 Dataset** The Covid-19 dataset [58] consists of Covid patients from Brazil. The dataset is publicly available and based on SIVEP-Gripe data [82]. The dataset consists of 6882 patients from Brazil recorded between February 27-May 4 2020. The dataset captures risk factors including comorbidities, symptoms, and demographic characteristics. There is a mortality label from Covid-19 making it a binary classification task. A summary of the characteristics of the covariates can be found in Table 7.

**SEER Dataset** The SEER dataset is a publicly available dataset consisting of 240,486 patients enrolled in the American SEER program [66]. The dataset consists of features used to characterize prostate cancer, including age, PSA (severity score), Gleason score, clinical stage, and treatments. A summary of the covariates can be found in Table 8. The classification task is to predict patient mortality, which is a binary label.

The dataset is highly imbalanced, where  94% of patients survive. Hence, we extract a balanced subset of 20,000 patients (i.e. 10,000 with label=0 and 10,000 with label=1).

**CUTRACT Dataset** The CUTRACT dataset is a private dataset consisting of 10,086 patients enrolled in the British Prostate Cancer UK program [67]. It includes the same features as SEER and also uses mortality as the label, see Table 8.

The dataset is highly imbalanced in its labels, hence we choose to extract a balanced subset of 2,000 patients (i.e. 1000 with label=0 and 1000 with label=1).

## B.2  Experiments

For specifics on how $G$ is evaluated, tuned, and selected, please see Appendix C.1.

Table 7: Summary of features for the Covid-19 Dataset [58]

| Feature | Range |
|---|---|
| Sex | 0 (Female), 1(Male) |
| Age | $1 - 104$ |
| Fever | $0, 1$ |
| Cough | $0, 1$ |
| Sore throad | $0, 1$ |
| Shortness of breath | $0, 1$ |
| Respiratory discomfort | $0, 1$ |
| SPO2 | $0 - 1$ |
| Diharea | $0, 1$ |
| Vomitting | $0, 1$ |
| Cardiovascular | $0, 1$ |
| Asthma | $0, 1$ |
| Diabetes | $0, 1$ |
| Pulmonary | $0, 1$ |
| Immunosuppresion | $0, 1$ |
| Obesity | $0, 1$ |
| Liver | $0, 1$ |
| Neurologic | $0, 1$ |
| Branca (Region) | $0, 1$ |
| Preta (Region) | $0, 1$ |
| Amarela (Region) | $0, 1$ |
| Parda (Region) | $0, 1$ |
| Indigena (Region) | $0, 1$ |

Table 8: Summary of features for the SEER [66] and CUTRACT [67] datasets. *Note: the range of age starts slightly lower for SEER (37-95) compared to CUTRACT (44-95).*

| Feature | Range |
|---|---|
| Age | $37 - 95$ |
| PSA | $0 - 98$ |
| Comorbidities | $0, 1, 2, \geq 3$ |
| Treatment | Hormone Therapy (PHT), Radical Therapy - RDx (RT-RDx),Radical Therapy -Sx (RT-Sx), CM |
| Grade | $1, 2, 3, 4, 5$ |
| Stage | $1, 2, 3, 4$ |
| Primary Gleason | $1, 2, 3, 4, 5$ |
| Secondary Gleason | $1, 2, 3, 4, 5$ |

### B.2.1 Experiment 5.1: Subgroups

In this experiment, we evaluate the performance estimates on different subgroups based on the mean absolute error compared to the estimates of subgroup performance using the oracle dataset. In order, to represent potential variation of selecting different test sets, we repeat the experiment 10 times, where we sample a different test set in each run. That being said, we keep the proportions in each dataset fixed such that $\{\mathcal{D}_{train,f}, \mathcal{D}_{test,f}, \mathcal{D}_{oracle}\} = \{8.4k, 2.1k, 19.6k\}$. Given that minority subgroups have few samples, in this experiment, we generate $n$ samples for each subgroup, where $n$ is the size of the largest subgroup in $\mathcal{D}_{test,f}$. This allows us to "balance" the evaluation dataset.

We provide more details on the subgroups below for each dataset:

In producing the intersectional performance matrix, we slice the data for these intersections. However, as we slice the data into finer intersections, the intersectional groups naturally become smaller. Hence, to ensure we have reliable estimates, we set a cut-off wherein we only evaluate performance for

Table 9: Details on dataset specific subgroups

| Dataset | Subgroup | Specifics |
|---|---|---|
| Adult | Race | " White", "Amer-Indian-Eskimo", "Asian-Pac-Islander", "Black", " Other" |
| Drug | Ethnicity | 'White', 'Other', 'Mixed-White/Black', 'Asian', 'Mixed-White/Asian', 'Black', 'Mixed-Black/Asian' |
| Covid | Region | 'Branca', 'Preta', 'Amarela', 'Parda', 'Indigena' |
| Support | Race | 'white','black', 'asian', 'hispanic' |
| Bank | Employment status (Anon) | 'CA', 'CB', 'CC', 'CD', 'CE', 'CF', 'CG' |

intersectional groups where there are 100 or more samples. In computing the mean absolute error, we do not include the corresponding intersections for which there were insufficient samples.

For Section 5.1.2, we use bootstrapping for the naive $\mathcal{D}_{test,f}$ baseline. Bootstrapping is a prevalent method for providing uncertainty [21, 62, 83–85]. In our case, we sample a dataset $\mathcal{D}_{test,f}^k$ the same size as $\mathcal{D}_{test,f}$, uniformly with replacement from $\mathcal{D}_{test,f}$. Model $f$ is evaluated on each $\mathcal{D}_{test,f}^k$, and the mean and standard deviation across $\mathcal{D}_{test,f}^k$ is used to construct confidence intervals. For all methods, intervals are chosen as the mean $\pm 2$ standard deviations.

### B.2.2 Experiments 5.2 Distributional shifts

**No prior knowledge: characterizing sensitivity across operating ranges.** In this experiment, we assess performance for different degrees of shift, we compute three shift buckets around the mean of the original feature distribution: large negative shift from the mean (**-**), small negative/positive shift from the mean (**±**), and large positive shift from the mean (**+**).

We define each in terms of the feature quantiles. We generate uniformly distributed shifts (between min(feature) and max(feature)). Any shift that shifts the mean to less than Q1 is (**-**), any shift that shifts the mean to more than Q3 is (**+**) and any shift in between is (**±**). The Oracle target is created using rejection sampling of the oracle source data, see Section B.3.

**Prior Knowledge.** In this experiment, we assume we observe some of the features in the target domain, i.e. we observe the empirical marginal distribution of $X_c$. This empirical marginal distribution is used to sample from, and conditioned on when generating the other features. The Source RS target is created using rejection sampling of the test data, see Section B.3.

### B.3 Rejection sampling for creating shifted datasets

In experiments 5.2 and 5.3, we use rejection sampling for the oracle and source baselines, respectively. Let us briefly explain how this is achieved.

Let $\mathcal{D}$ be some dataset with distribution $p^0(X)$ that can be split into parts $p(X_{\bar{c}}|X_c)$ and $p(X_c)$. As noted in Section 3, we assume the latter changes (inducing a shift in distribution), while the former is fixed. We denote the shifted marginal distribution as $p^s(X_c)$ and the full distribution as $p^s(X) = p^s(X_c)p(X_{\bar{c}}|X_c)$.

In experiment 5.2, we desire a ground-truth target dataset for a given shift. We do not have data from $p^s(X)$, however we can use rejection sampling to *create* such dataset, which we denote by $\mathcal{D}^s$. Since we do not know $p^0(X)$ either, we sample from the empirical distribution, i.e. from data $\mathcal{D}$ itself, which will converge to the true distribution when $|\mathcal{D}|$ becomes large. To approximate $p^0(X_c)$, we train a simple KDE model and $p^s(X_c)$ is defined by shifting this distribution (see Section 4.2). This gives the following algorithm:

---

**Algorithm 1** Rejection sampling from source dataset $\mathcal{D}$, given a predefined marginal shift $T$ and desired test set size.

---

**Input** Source dataset $\mathcal{D}$, shift $T$ and desired shifted set size $n_s$
Fit density model $\hat{p}^0(X_c)$ to $\{\mathbf{x_c}|\mathbf{x} \in \mathcal{D}\}$
$\hat{p}^s(X_c) \leftarrow T(\hat{p}^0)(X_c)$
$M \leftarrow \max_{\mathbf{x_c} \in \mathcal{D}} \frac{\hat{p}^s(\mathbf{x_c})}{\hat{p}^0(\mathbf{x_c})}$
$\mathcal{D}^s \leftarrow \{\}$
**while** $|\mathcal{D}^s| < n_s$ **do**
    Sample $\mathbf{x}$ from $\mathcal{D}$ uniformly
    Sample $u \sim U(0,1)$
    **if** $\frac{\hat{p}^s(\mathbf{x_c})}{\hat{p}^0(\mathbf{x_c})} > Mu$ **then** $\mathcal{D}^s \leftarrow \mathcal{D}^s \cup \{\mathbf{x}\}$
    **end if**
**end while**
**return** $\mathcal{D}^s$

---

In Experiment 5.2 we run the above with $\mathcal{D}$ an oracle test set. Since the oracle test set is very large, it covers $p^0$ relatively well. This allows us to approximate $p^0(X_c)$, and also means that draws from the empirical distribution are distributed approximately like the true underlying distribution.

In experiment 5.3 we use a similar set-up for creating baseline *Source (RS)* based on $\mathcal{D}_{test,f}$ alone, and to have a fair comparison we use rejection sampling to weigh the points.[7] In this case, however, the distribution $p^0(X_c)$, and in turn $p^s(X_c)$, cannot be approximated accurately. In addition, we may have very little data such that the same points need to be included many times (in regions with large $p^s(X_c)/p^0(X_c)$). As a result, although we see that *Source (RS)* performs better than unshifted *Source (all)*, it is a poor evaluation approach.

---

[7]Effectively, this reduces to an importance weighted estimate of the performance.

# C   Generative Model Choice

Any generative model can be used to produce the synthetic test data, but some models may be better or worse than others. 3S uses CTGAN [47] since this model is designed specifically for tabular data and has shown good performance. In this section, we explain how it is tuned and include a comparison to other generative models.

## C.1   Assessing the quality of generative model G in 3S.

Approaches to model selection and quality assessment of generative models often measure the distance between the generated and the true distributions [86]. In 3S , we use Maximum Mean Discrepancy (MMD) [56], a popular choice for synthetic data quality [87, 88].

MMD performs a statistical test on distributions $P^r$ (Real) and $P^g$ (Generated), measuring the difference of their expected function values, with a lower MMD implying $P^g$ is closer to $P^r$.

We use MMD in our auto-tuning and model selection step, comparing the generated data to a held-out test set, with $G$ selected as the model with the lowest MMD. This step also serves to ensure that the data generated by 3S is indeed close to the real-world reference dataset of interest. Specifically, hyper-parameters of 3S when training $G$ are tuned via a Tree-structured Parzen Estimator. We search over the number of epochs of training [100, 200, 300, 500], learning rate [2e-4, 2e-5, 2e-6], and embedding dimension [64,128,256]. For all methods, we have a small hyper-parameter validation set with a size of 10% of the training dataset. Our objective is based on MMD minimization.

That said, of course, alternative widely used metrics such as Inverse KL-Divergence or the Jensen-Shannon divergence could also be used as metrics of assessment.

## C.2   Influence of model choice.

Any generative model can be used as the core of 3S . For efficiency, a conditional generative model is highly desirable; this allows direct conditioning on subgroup or shift information, and not e.g. post-generation rejection sampling. Furthermore, some generative models may provide more or less realistic data. Here we compare 3S estimates provided by CTGAN, vs estimates given by TVAE and Normalizing Flows.

We assess these different base models for $G$ on the race subgroup task from Sec. 5.1. Using 3S can assess the generative models based on MMD, but for completeness we also show inverse KL-divergence and Jensen-Shannon Divergence (JSD), where the metrics are computed vs a held-out validation dataset.

We show in Table 10, that the better quality metric does indeed translate into better performance when we use the synthetic data for model evaluation. We find specifically that CTGAN outperforms the other approaches, serving as validation for our selection.

Additionally, the results highlight that for practical application, one could evaluate the quality of $G$ first using metrics such as MMD, inverse KLD, or JSD, as a proxy for how well the generative model should perform.

We assume for the purposes of this experiment that the three classes of models are trained with the same optimization hyperparameters (epochs=200, learning rate=2e-4).

**Why do small changes in metrics, lead to large MAE differences?** One might wonder that small changes in for example JSD lead to large MAE. For instance, CTGAN to NF of 0.03 to 0.09. We assess the sensitivity of performance estimates to small changes in divergence measures.

To do so, we conduct an experiment where we synthetically increase the JSD of the synthetic dataset through corruption. We then compute the MAE performance estimate on the synthetic dataset compared to the Oracle for different JSD values.

The result in Figure 7 shows that even small changes in the JSD can significantly harm the MAE. This provides an explanation of why a method with seemingly relatively similar JSD (or divergence metric), with only a minor difference, may perform very differently as measured by MAE. This motivates why we select the generative model, for example with the lowest metric despite them looking similar.

Table 10: Assessing the influence of model choice for $G$ and illustrating how our quality assessment metrics in 3S can be used to select the best model which indeed will provide the best performance. We see that indeed CTGAN performs best in this case.

| Base $G$ | MMD $\downarrow$ | Inverse KLD $\uparrow$ | JSD $\downarrow$ | Subgroup (%) | Mean Absolute Error % $\downarrow$ |
|---|---|---|---|---|---|
| CTGAN | **0.0014** | **0.995** | **0.03** | #1 (86%) | **$0.28 \pm 0.24$** |
| | | | | #2 (9%) | **$17.64 \pm 0.29$** |
| | | | | #3 (3%) | **$2.96 \pm 1.02$** |
| | | | | #4 (1%) | **$1.14 \pm 0.62$** |
| | | | | #5 (1%) | **$1.03 \pm 0.85$** |
| NF | 0.0034 | 0.970 | 0.09 | #1 (86%) | $16.25 \pm 0.53$ |
| | | | | #2 (9%) | $26.35 \pm 1.07$ |
| | | | | #3 (3%) | $20.50 \pm 3.31$ |
| | | | | #4 (1%) | $27.14 \pm 0.97$ |
| | | | | #5 (1%) | $26.04 \pm 2.87$ |
| TVAE | 0.4557 | 0.4987 | 0.505 | #1 (86%) | $25.93 \pm 1.34$ |
| | | | | #2 (9%) | $35.40 \pm 0.83$ |
| | | | | #3 (3%) | $24.05 \pm 1.12$ |
| | | | | #4 (1%) | $33.0 \pm 0.31$ |
| | | | | #5 (1%) | $37.69 \pm 0.61$ |

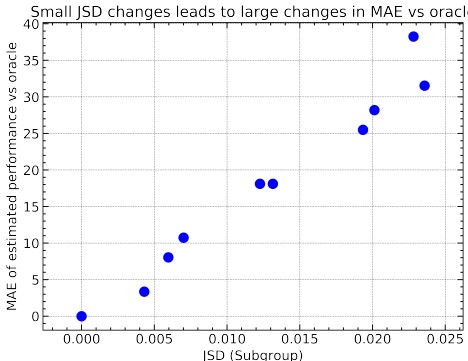

Figure 7: Small changes in JSD lead to large changes in MAE vs oracle.

———————————————

# D  Limitations and Failure Cases

## D.1  Limitations

We summarise 3S's main limitations (where applicable referencing limitations already outlined in the paper).

1. **Subgroups in real test set are large and 3S is unnecessary.** When the aim is subgroup evaluation and there is sufficient real data for this subgroup, the real data estimate on $\mathcal{D}_{test,f}$ will be sufficiently accurate and there is no need to use 3S given the computational overhead. See*Failure Case 1 for new experiments and further discussion. Note, even for very large datasets, there can be very sparse regions or small subgroups for which using 3S is still beneficial. Also note that this limitation mostly applies to subgroup evaluation and less to Generating Synthetic Data with Shifts, because performance estimates for possible shifts is less trivial using real data alone (e.g. require reweighting or resampling of test data) and we show (Sec. 5.2, Table 1 and Fig. 6b) that 3S beats real data baselines consistently.

2. **Possible errors in the generative process**. Errors in the generative process can affect downstream evaluation. This is especially relevant for groups or regions with few samples, as it is more likely a generative model does not fit the distribution perfectly here. By replacing the generative model by a deep generative ensemble [25], 3S provides insight into its own generative uncertainty. When there is very little data and 3S's uncertainty bounds are too large, practitioners should consider gathering additional data. See Failure Case 2 below.

3. **Not enough knowledge or data to generate realistic shifts.** The success of modelling data with distributional shifts relies on the validity of the distributional shift's assumptions. This is true for 3S as much as for supervised works on distributional shifts (e.g. see [89]). In Sec. 5.2.2 we show how a lack of shift knowledge may affect evaluation. We aimed to generate data to test a model on UK patients, using mostly data from US patients. We do not define the shift explicitly—we only assume that we observe a small number (1 to 4) of features $X_c$ (hence the conditional distribution of the rest of the features conditional on these features is the same across countries). In Fig. 6b, we show the assumption does not hold when we only see one feature—this lack of shift knowledge is a failure case. When we add more features, the assumption is more likely to approximately hold and 3S converges to a better estimate. We reiterate that invalid shift assumptions are a limitation of any distributional shift method, e.g. the rejection sampling (RS) baseline using only real data is worse than 3S overall.

4. **Computational cost**. The computational cost and complexity of using generative models is always higher than using real test data directly. For typical tabular datasets, the computational requirements are often very manageable: under 5 min for most datasets and under 40 min for the largest dataset Bank (Table 11). The reported times correspond to the dataset sizes in the paper. We would like to emphasise that the cost at deployment time for a poorly evaluated model can be unexpectedly high, which warrants 3S's higher evaluation cost. Additionally, pre-implemented generative libraries (e.g. Patki 2016, Qian 2023) can accelerate the generative process, automating generator training with minimal user input.

Table 11: Training time for generator on an NVIDIA GeForce RTX3080 for main paper settings.

| Dataset | Adult | Covid | Drug | Support | Bank |
|---|---|---|---|---|---|
| Tuning time (min) | 4.5 | 4 | 1.5 | 2 | 38 |

## D.2  Failure cases

For limitations 1 and 2 mentioned above, we include two new experiments highlighting failure cases on two extreme settings:

1. **3S is similarly accurate to real data evaluation when the real test data is large.** With sufficiently large real data, 3S provides no improvement despite higher complexity. We can determine sufficient size by estimating the variance $Var(A)$ of the performance metric $A(f; D, S)$ w.r.t. *the random variable* denoting the test data $D$. With only access to one real dataset $D_{test}$ however, we can approximate $Var(A)$ via bootstrapping (Fig. 5). If $Var(A)$

falls below a small threshold $\alpha$ (e.g. 0.05), practitioners may for example decide to trust their real data estimate and not use 3S, as further evaluation improvements are unlikely. In our Bank dataset experiment (500k examples), we vary age and credit risk thresholds, retaining samples above each cut-off to shrink the test set (see Figure 9). Note that for large datasets but very small subgroups, the $D_{test,f}$ estimate still has a high variance, (reflected in the large bootstrapped $Var(A)$ ), hence this should urge a practitioner to use 3S.

2. **Large uncertainty for very small test sets.** At the other extreme, when there are too few samples the uncertainty of 3S (quantified through a DGE [25]) can become too large. In Figure 8 in which we reduce subgroups to fewer than 10 samples in the test data. We train $G$ in 3S on the overall test set (which includes the small subgroup) with $n_{samples}$. Despite good performance versus an oracle, the uncertainty intervals from 3S's ensemble span 0.1-0.2. These wide intervals make the 3S estimates unreliable and less useful, and would urge a practitioner to consider gathering additional data. The key takeaway: With extremely sparse subgroups, the large uncertainties signal that more data should be gathered before relying on 3S's uncertain estimates.

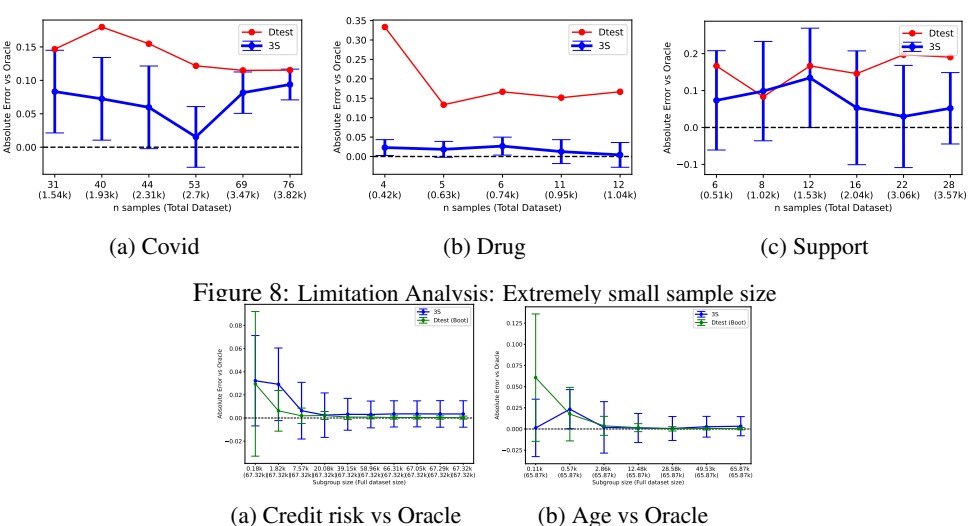

(a) Covid        (b) Drug        (c) Support

Figure 8: Limitation Analysis: Extremely small sample size

(a) Credit risk vs Oracle        (b) Age vs Oracle

Figure 9: Limitation analysis: Large sample size

# E  Additional Experiments

This appendix includes a number of additional results. First, we include additional results for the main paper's experiments, using different downstream models, predictors, metrics, and baselines. Second, we discuss other types of subgroup definitions, including subgroups based on points of interest and density of regions. Third, we include results for 3S when we have access to the training set of predictor model $f$; since this dataset is usually larger, it can help train a more accurate generative model.

## E.1  Experimental additions to main paper results

### E.1.1  Sanity check

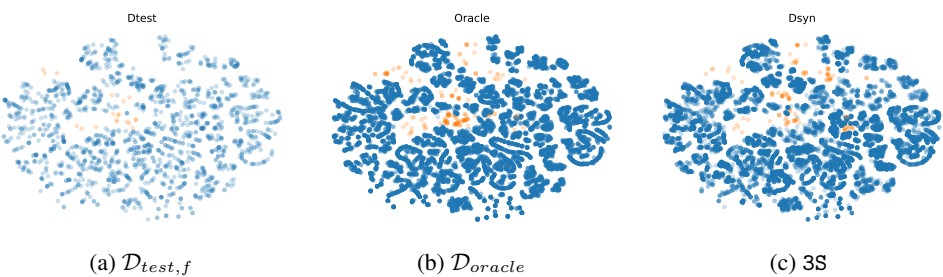

(a) $\mathcal{D}_{test,f}$           (b) $\mathcal{D}_{oracle}$           (c) 3S

Figure 10: Qualitative assessment of synthetic data. T-SNE on the Adult dataset comparing the real test data $\mathcal{D}_{test,f}$, oracle data $\mathcal{D}_{oracle}$ and 3S data $\mathcal{D}_{syn}$. We find that 3S generates synthetic test data that covers the oracle dataset well, despite only having access to $\mathcal{D}_{test,f}$ during training. The data, evaluating subgroups where blue (Race 1 - majority) and orange (Race 5 - minority)

---

We perform a sanity check on the data generated by 3S as compared to $\mathcal{D}_{oracle}$ and $D_{test}$. The results are visualized in Figure 10 and show that 3S generates synthetic test data that covers the oracle dataset well, even on the minority subgroup in blue, capturing the distribution appropriately.

### E.1.2  Subgroup performance evaluation: more metrics and downstream models

**Motivation.** The performance of the model, on subgroups, is likely influenced by the class of downstream predictive model $f$. We aim to assess the performance of the granular subgroups for a broader class of downstream models $f$. In addition, to accuracy, we assess F1-score performance estimates as another showcase of downstream performance estimates.

**Setup.** This experiment evaluates the mean performance difference for (i) 3S , (ii) 3S +, and (iii) $\mathcal{D}_{test,f}$. We follow the same setup as the granular subgroup experiment in Section 5.1. We increase the predictive models beyond RF, MLP, and GBDT and further include SVM, AdaBoost, Bagging Classifier, and Logistic Regression.

**Analysis.** Figures 11 and 12 illustrate that 3S , when evaluated with more models, still better approximates the true performance on minority subgroups, compared to test data alone. This is in terms of mean absolute performance difference between predicted performance and performance evaluated by the oracle.

**See next page for results figures**

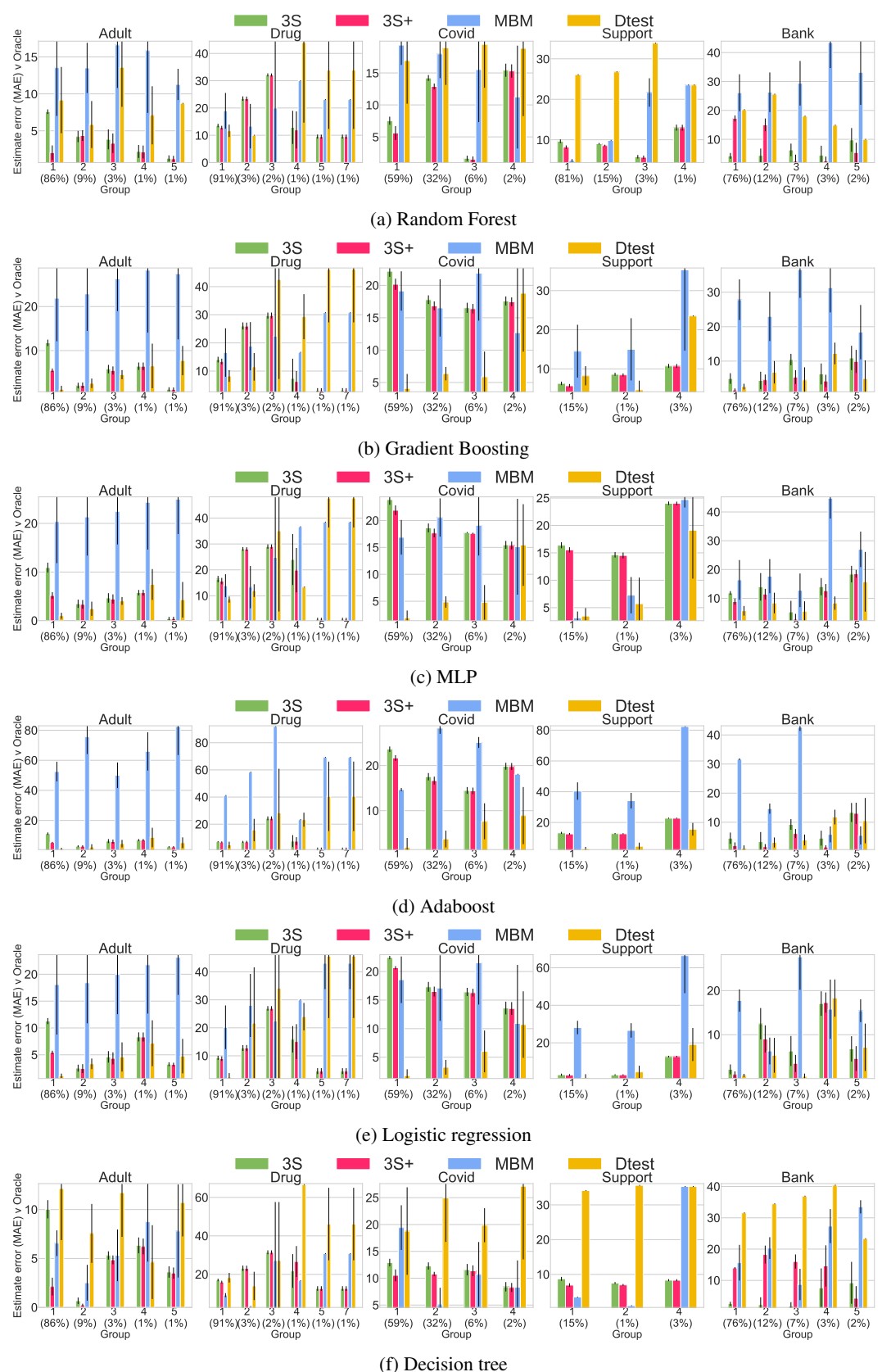

Figure 12: Performance estimate error wrt an Oracle: Accuracy

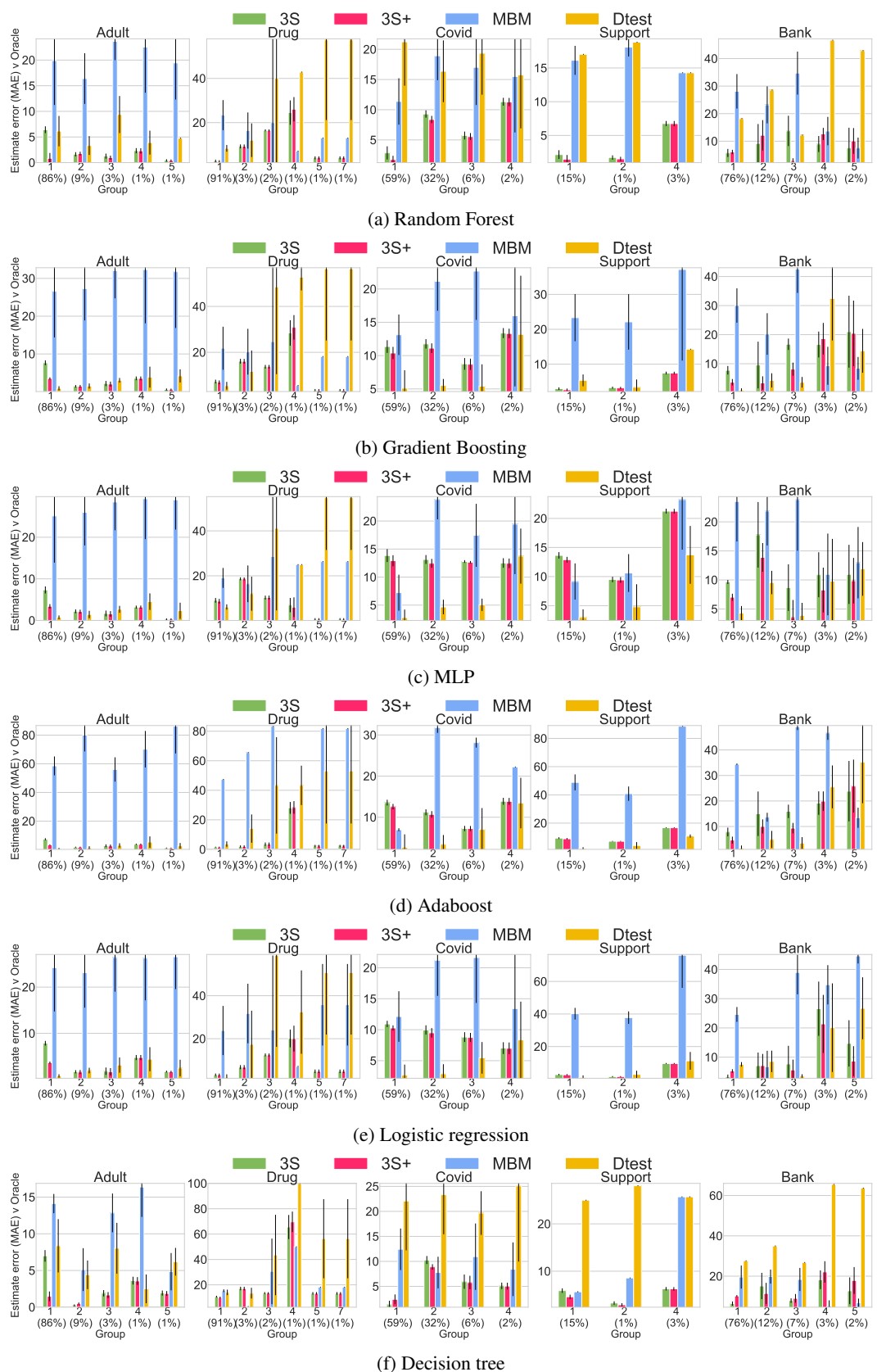

Figure 11: Performance estimate error wrt an Oracle: F1-score

### E.1.3 Subgroup worst-case performance evaluation

**Motivation.** When estimating sub-group performance, of course we want to have as low error as possible on average (i.e. low mean performance difference). That said, average performance glosses over the worst-case scenario. We desire that the worst-case mean performance difference is also low. This is to ensure that, by chance, the performance estimates are not wildly inaccurate. This scenario is particularly relevant, as by chance the testing data could either over- or under-estimate model performance, leading us to draw incorrect conclusions.

**Setup.** This experiment evaluates the worst-case mean performance difference for (i) 3S , (ii) 3S +, and (iii) $\mathcal{D}_{test,f}$. We follow the same setup as the granular subgroup experiment in Section 5.1.

**Analysis.** Figures 13 and 14 illustrates that 3S and the augmented 3S + have a lower worst-case performance compared to evaluation with real test data. This further shows that, by chance, evaluation with real data can severely over- or under-estimate performance, leading to incorrect conclusions about the model's abilities. 3S 's lower worst-case error, means even in the worst scenario, that 3S 's estimates are still closer to true performance.

**See next page for results figures**

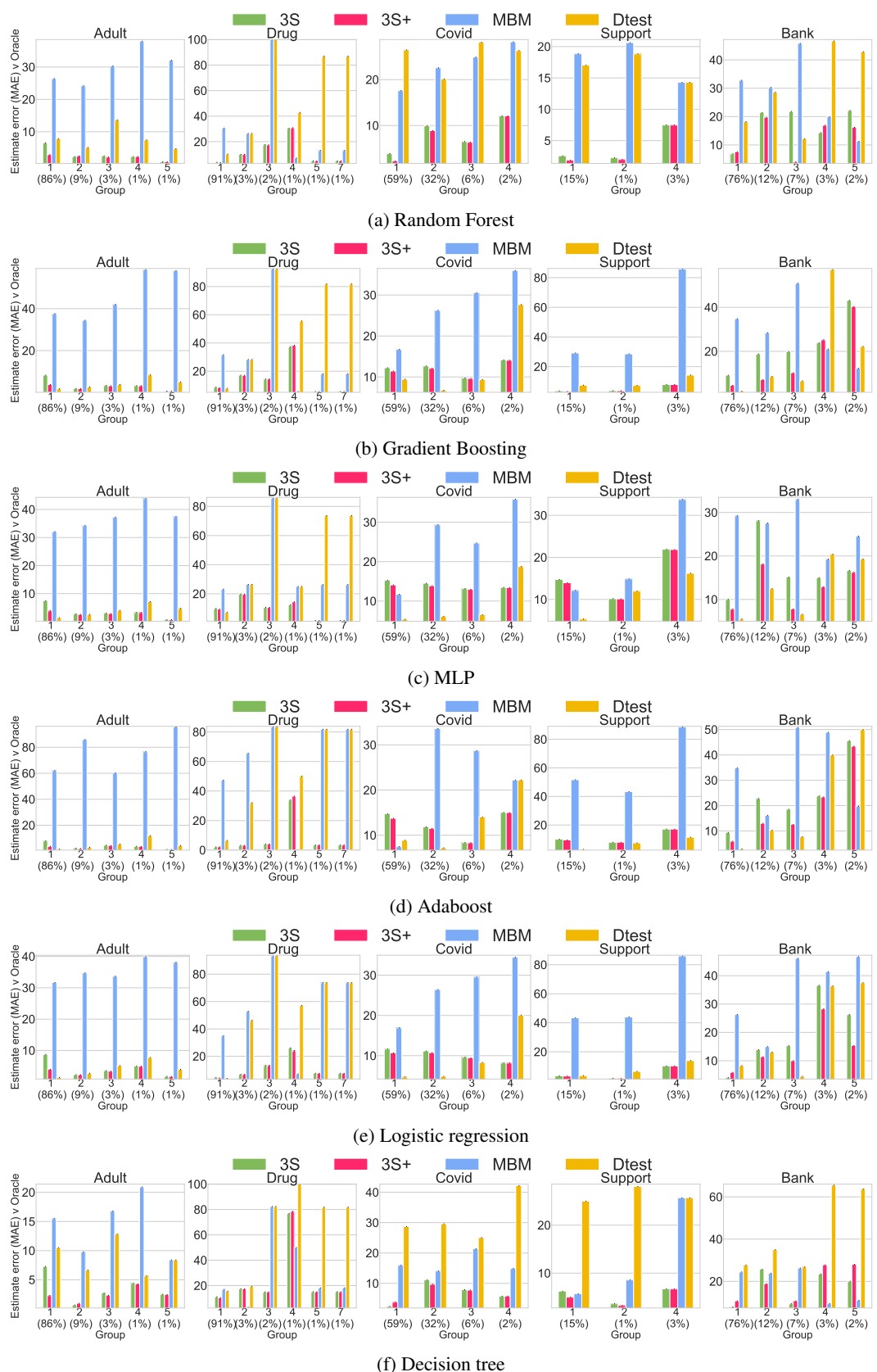

Figure 13: Worst case, Performance estimate error wrt an Oracle: F1 Score

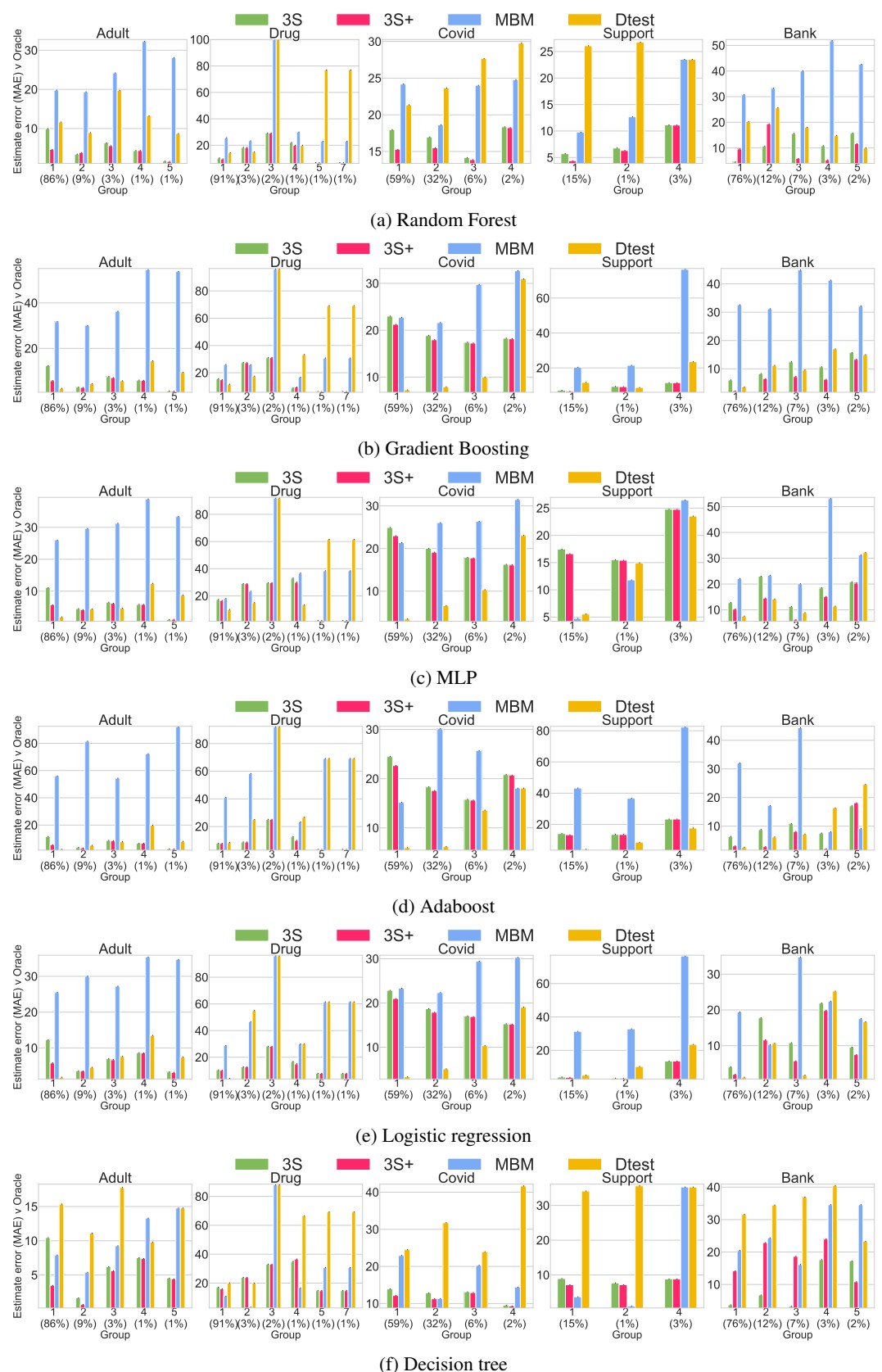

Figure 14: Worst case, Performance estimate error wrt an Oracle: Accuracy

### E.1.4 Fairness metrics: subgroup evaluation

**Motivation.** We have primarily studied reliable estimation of model performance on different subgroups. This is easily generalized to estimate fairness metrics of ML models on specific subgroups. This can provide further insight into the use of synthetic data for model testing.

**Setup.** This experiment evaluates the mean performance difference for (i) 3S , (ii) 3S +, and (iii) $\mathcal{D}_{test,f}$. We follow the same setup as the granular subgroup experiment in Section 5.1. We evaluate an RF model. We assess the following fairness metrics: (i) Disparate Impact (DI) ratio (demographic parity ratio) and (ii) Equalized-Odds (EO) ratio. When estimating these metrics for each subgroup (e.g. race group), we then condition on sex as the sensitive attribute.

The DI ratio is: the ratio between the smallest and the largest group-level selection rate $E[f(X)|A = a]$ , across all values of the sensitive feature(s) $a \in A$.

The EO ratio is the smaller of two metrics between TPR ratio (smallest and largest of $P[f(X) = 1|A = a, Y = 1]$ , across all values of the sensitive feature(s)) and FPR ratio (similar but defined for $P[f(X) = 1|A = a, Y = 0]$), , across all values of the sensitive feature(s) $a \in A$.

**Analysis.** Figures 15 and 16 illustrate that 3S 's performance on both fairness metrics, better approximates the true oracle metric on minority subgroups, compared to test data alone. This is in terms of mean absolute performance difference between predicted performance and performance evaluated by the oracle.

Note, we also assess the worst-case scenario as well, as done previously.

Figures 17 and 18 illustrate that 3S and the augmented 3S + have a lower worst-case estimated difference compared to evaluation with real test data. This further shows that, by chance, evaluation with real data can over- or under-estimate fairness, leading to incorrect conclusions about the model's abilities. 3S 's lower worst-case error, means even in the worst scenario, that 3S 's estimates are still closer to the true fairness metric.

**See next page for results figures**

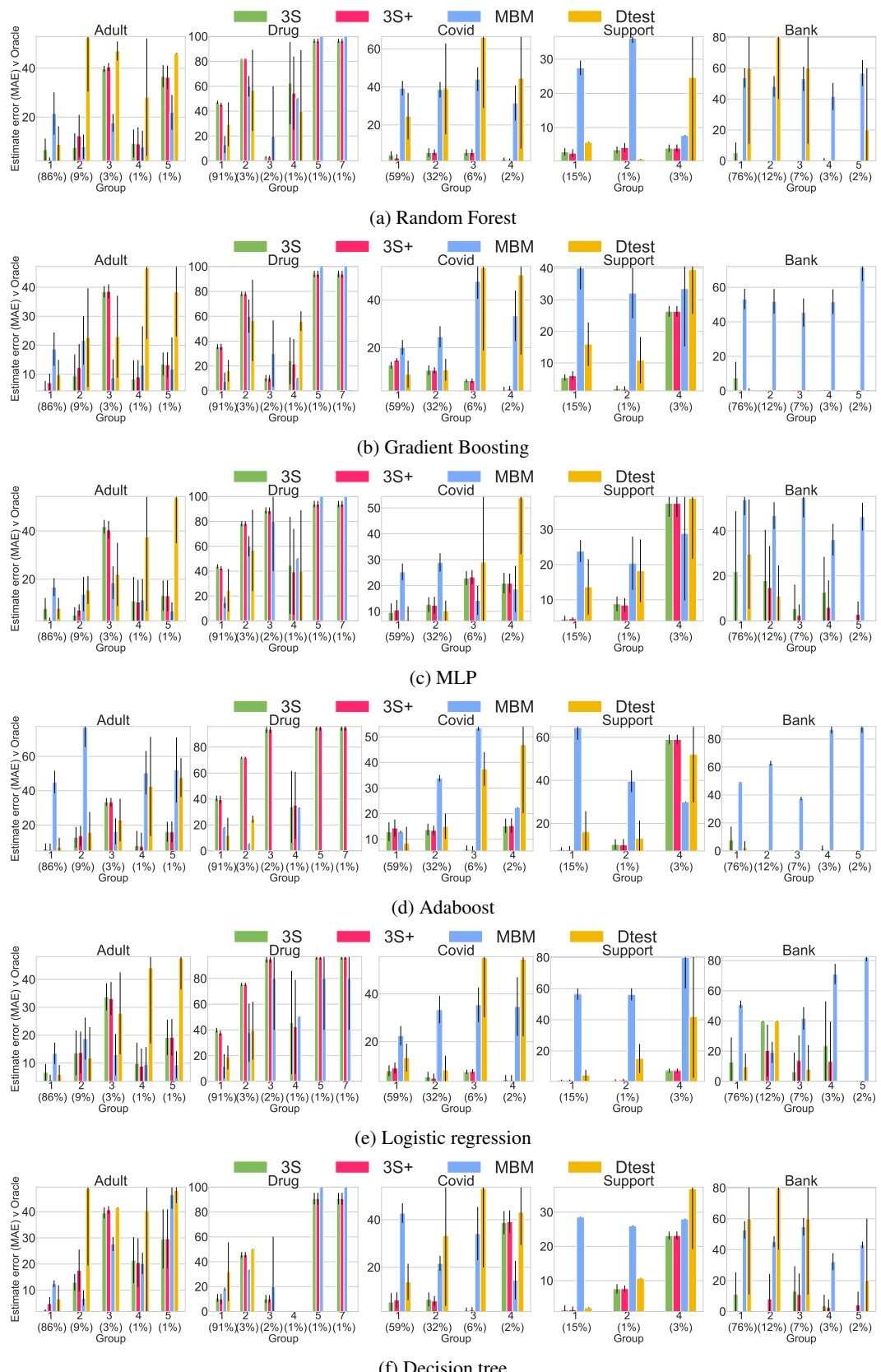

Figure 15: Fairness (in terms of EO) estimate error

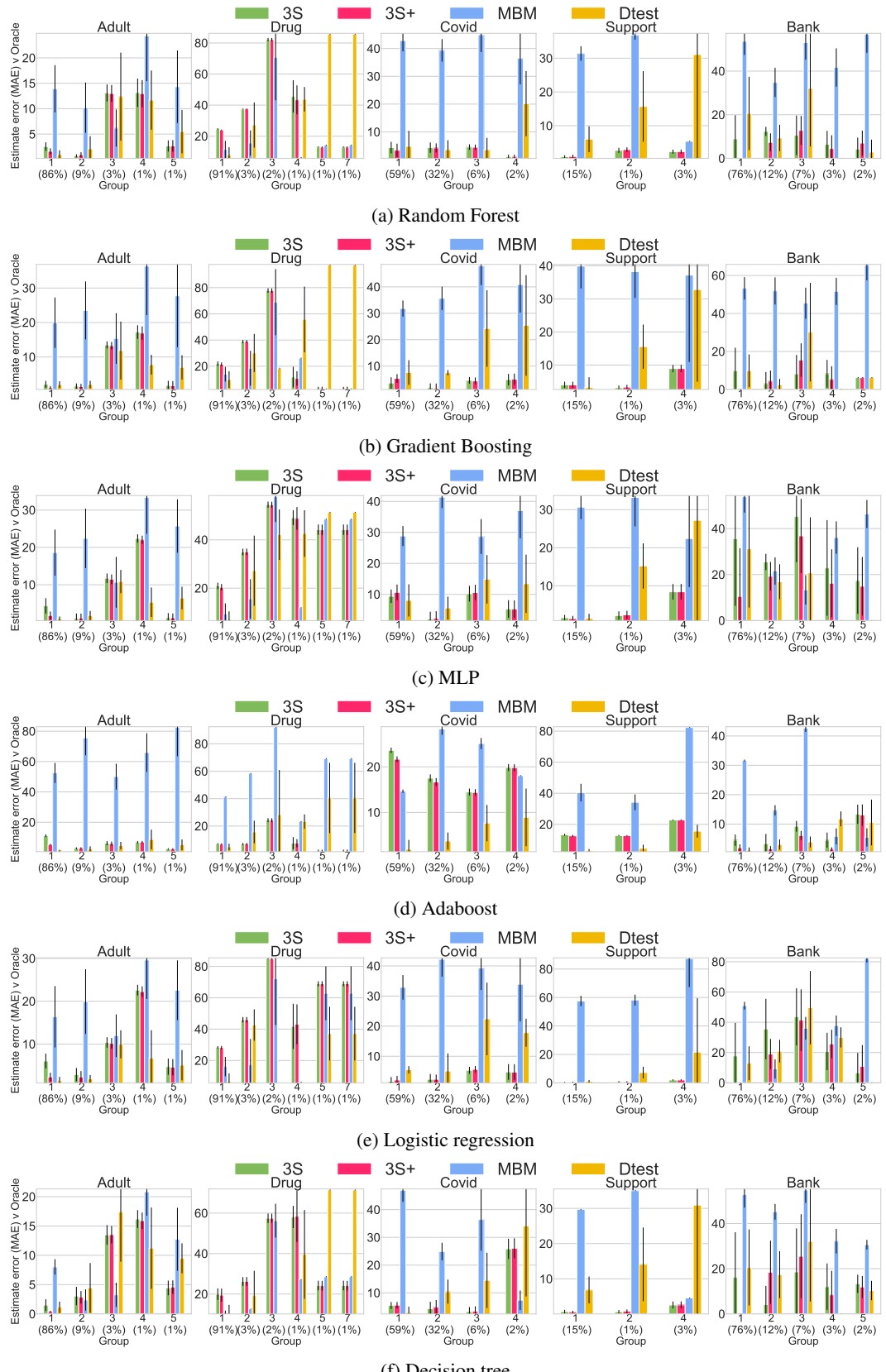

Figure 16: DI estimate error

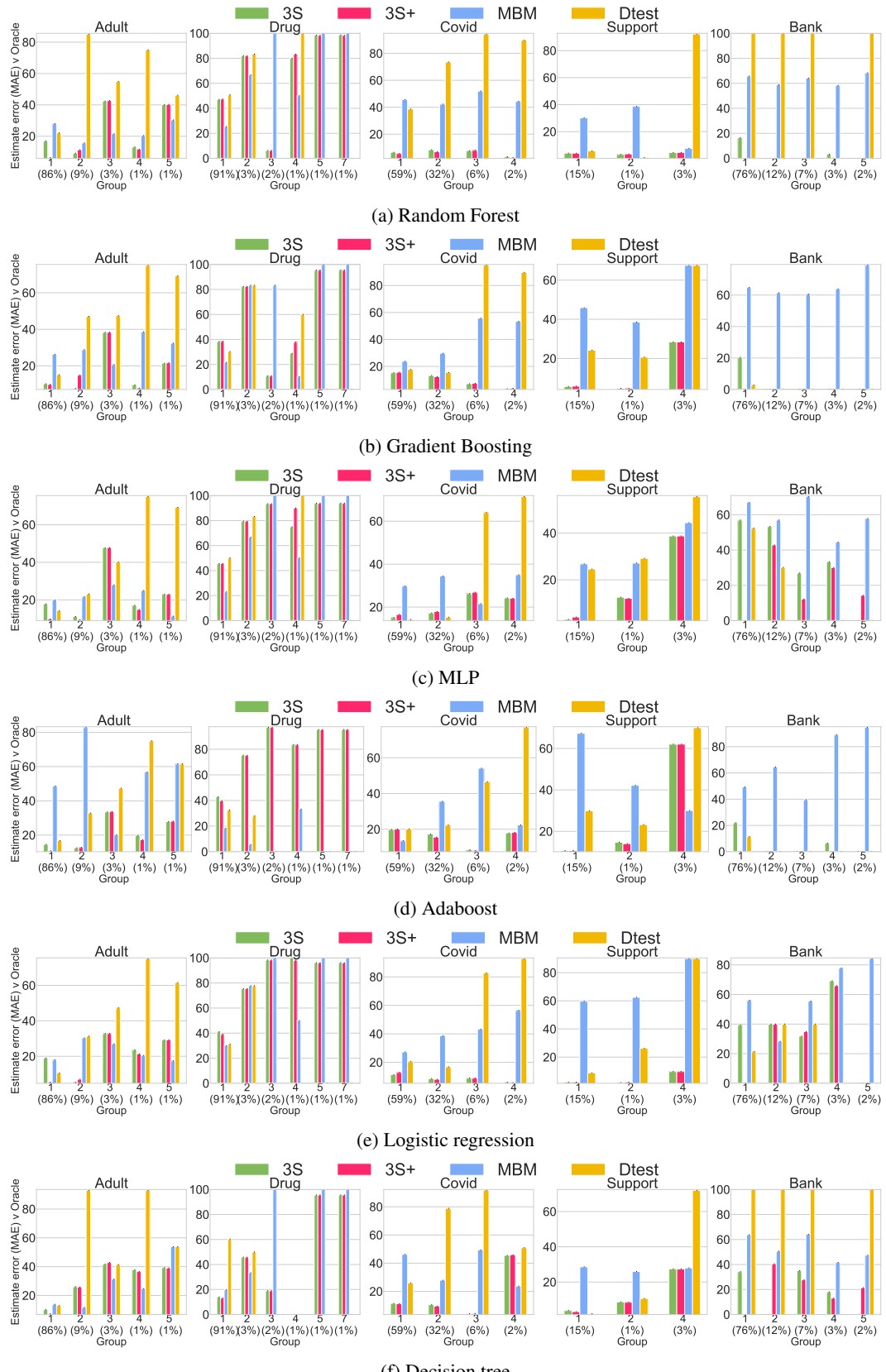

Figure 17: Worst case, EO estimate error

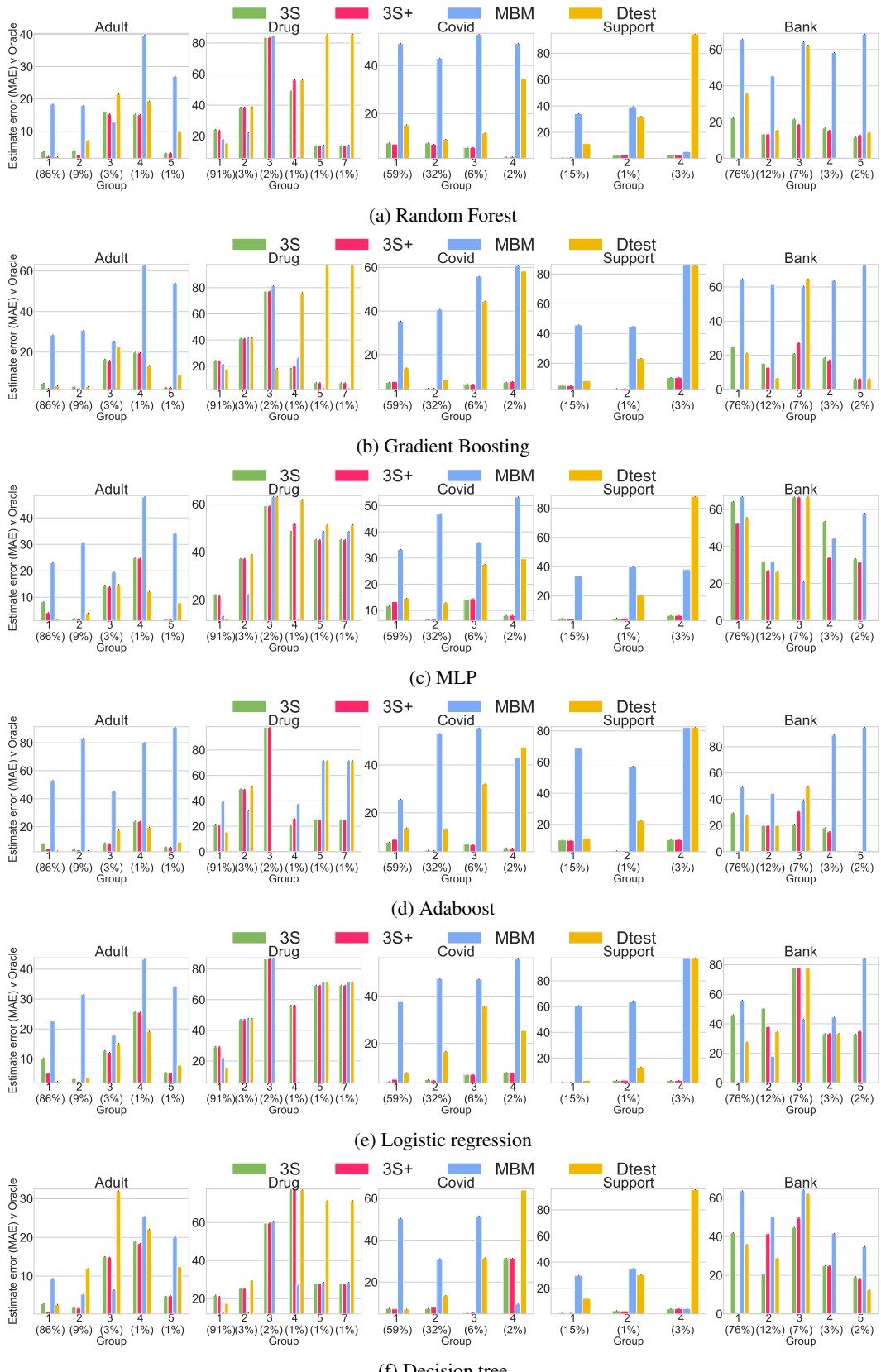

Figure 18: Worst case, DI estimate error

### E.1.5 Intersectional subgroups: performance matrix

**Motivation.** We perform a deep-dive of the intersectional performance matrices generated by 3S , simply, using test data alone and the oracle.

**Analysis.** We first see in Table 12 that, in general, the intersectional matrix 3S has a much *lower* error when estimating performance for intersectional subgroups. i.e. this is more similar to the oracle, compared to using $\mathcal{D}_{test,f}$. Note that we set the minimum number of samples required for validation $= 100$ samples. This induces sparseness, of course, but is necessary in order to prevent evaluation on too few data points. However, in cases where $\mathcal{D}_{test,f}$ does not have data for the intersection (i.e. $n < 100$), we do not consider these NaN blocks as part of our calculation; in fact, this makes it easier for $\mathcal{D}_{test,f}$.

The rationale is evident when evaluating the intersectional performance matrices for each group. We present the following findings.

- 3S **'s insights are correct**: the underperforming subgroups, as noted by 3S , match the oracle. Therefore, it serves as further validation.

- $\mathcal{D}_{test,f}$ **is very sparse after cut-offs**: the 100 sample cut-off highlights the key challenge of evaluation on a test set. We may not have sufficient samples for each intersection to perform an evaluation.

Table 12: Adult: Intersectional performance matrix difference vs the oracle

| Model | 3S | $\mathcal{D}_{test,f}$ |
|---|---|---|
| Average | $0.13 \pm 0.005$ | $0.21 \pm 0.002$ |
| RF | 0.133 | 0.211 |
| GBDT | 0.128 | 0.207 |
| MLP | 0.128 | 0.207 |
| SVN | 0.138 | 0.209 |
| AdaBoost | 0.126 | 0.206 |
| Bagging | 0.138 | 0.211 |
| LR | 0.126 | 0.207 |

### E.1.6 Incorporating prior knowledge on shift: Covid-19

**Motivation.** We have the ability to assess distributional shift where we have *some* knowledge of the shifted distribution. Specifically, here we assume we only observe a few of the features, from the target domain.

**Setup.** Our setup is similar to Section 5.2.2, however on a different dataset - i.e. Covid-19. There are known distributional differences between the north and south of Brazil. For example, different prevalence of respiratory issues, sex proportions, obesity rates, etc. Hence we train the predictive model on patients from the South (larger population) and seek to evaluate potential performance on patients from the North. We take the largest sub-regions for each.

To validate our estimate, we use the actual northern dataset (Target) as ground-truth. Our baselines are as in Section 5.2.2. Since the features are primarily binary, we parameterize the distributions as binomial with a probability of prevalence for their features. We can then sample from this distribution.

**Analysis.** Fig. 19 shows the average estimated performance of $f$, as a function of the number of features observed from the target dataset. We see that the 3S estimates are closer to the oracle across the board compared to baselines. Furthermore, for increasing numbers of features (i.e. increasing prior knowledge), we observe that 3S estimates converge to the oracle.

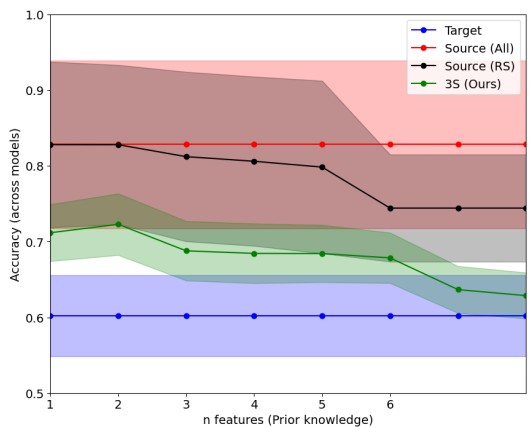

Figure 19: $\mathcal{D}_{syn}$ is better able to approximate performance in the target domain compared to baselines and that performance improves as more prior knowledge is incorporated via added features. Points are connected to highlight trends.

**Raw performance.** We also illustrate raw estimate errors for different downstream models in Table 13, which illustrates that 3S outperforms real data alone, rejection sampling, as well as, specialized methods ATC, IM and DOC for OOD prediction with lower downstream performance estimate errors.

Table 13: 3S has lower performance estimate error in target domain for different downstream models. Rows yellow have access to more information than 3S, in the form of unlabeled data from the target domain. ↓ is better

|  | mean | ada | bag | gbc | mlp | rf | knn | lr | knn | dt |
|---|---|---|---|---|---|---|---|---|---|---|
| 3S-Testing | **0.0446** | **0.0112** | **0.0962** | **0.0205** | 0.1777 | **0.0278** | **0.0100** | **0.0105** | **0.0100** | **0.0100** |
| All (Source) | 0.2262 | 0.1029 | 0.3890 | 0.0929 | 0.2276 | 0.3732 | 0.1774 | 0.0521 | 0.1774 | 0.3942 |
| RS (Source) | 0.1421 | 0.1842 | 0.1781 | 0.1303 | 0.2954 | 0.1613 | 0.0497 | 0.1138 | 0.0497 | 0.0244 |
| ATC [31] | 0.1898 | 0.0236 | 0.2948 | 0.0542 | **0.0331** | 0.2264 | 0.3985 | 0.0638 | 0.3985 | 0.4198 |
| IM [33] | 0.1782 | 0.1030 | 0.3638 | 0.0330 | 0.1563 | 0.2828 | 0.0555 | 0.0140 | 0.0555 | 0.4170 |
| DOC [32] | 0.1634 | 0.1019 | 0.2646 | 0.0304 | 0.1557 | 0.2215 | 0.1186 | 0.0010 | 0.1186 | 0.4060 |

#### E.1.7 Sensitivity to shifts on additional features

**Motivation.** In the main paper, we have characterized the sensitivity across operating ranges for two features in two datasets (Adult and SEER). Ideally, a practitioner would like to understand the sensitivity to all features in the dataset. We now conduct this assessment on the Adult dataset, producing model sensitivity curves for all features.

**Analysis.** We include the model sensitivity curves for all features as part of the example model report in Appendix F, see Figures 22, (a)-(t).

## E.2 Other definitions of subgroups

We illustrate how the definition of subgroup allows two other interesting use cases for granular evaluation: (i) subgroups defined using a point of interest (i.e. how would a model perform on patients that look like X), and (ii) subgroups defined in terms of local density (i.e. how would a model perform on unlikely patients).

### E.2.1 Subgroups relating to points of interest

**Motivation.** End-users may also be interested in knowing how a model performs on some point of interest $x^*$. For example, a clinician may have access to a number of models and may need to decide for the specific patient in front of them what the model's potential predictive performance might be for the specific patient. This is relevant because global performance metrics may hide that models underperform for specific samples.

We can often assess model performance on samples similar to the sample of interest, i.e. "neighborhood performance". A challenge in reality is that we often only have access to a held-out test dataset (i.e. $\mathcal{D}_{test}$), rather than the entire population (i.e. $\mathcal{D}_{oracle}$). How then can we quantify performance?

Usually, it is not possible to assess local performance using test data alone, since there may be very few samples that are similar enough—with "similar" defined in terms of some distance metric, i.e. $\mathcal{S} = \{x \in \mathcal{X} | d(x, x^*) < \epsilon\}$, with distance metric $d$ and some small distance $\epsilon \geq 0$. As before, we can instead generate synthetic data in the region $\mathcal{S}$ and compute the performance on this set instead. This reduces the dependence on the small number of samples, in turn reducing variance in the estimated performance.

**Set-up.** We compare two approaches: (i) find nearest-neighbor points in $\mathcal{D}_{test}$)—which might suffer from limited similar samples—, or (ii) use 3S and generate synthetic samples $\mathcal{D}_{syn}$ in some neighborhood of $x^*$. Similarly as before, we assess these two methods by comparing estimates with a pseudo-ground truth that uses nearest neighbors on a much larger hold-out set, $\mathcal{D}_{oracle}$. Again, we compare (1) *mean absolute performance difference* and (2) *worst-case performance difference* between a specific evaluation set and the oracle dataset. We average across 10 randomly queried points $x^*$ and use $k = 10$ nearest neighbors.

**Analysis.** The results in Table 14 show that 3S has a much lower neighborhood performance gap, both average and worst case for $x^*$ across models, when compared to the assessment using $\mathcal{D}_{test}$. The rationale is that by using synthetic data, we can generate more examples $\mathcal{D}_{syn}$ that closely resemble $x^*$, whereas with $\mathcal{D}_{test}$ we might be limited in the similar samples that can be queried - hence resulting in the higher variance estimates and poorer overall performance.

Table 14: Comparing two types of query methods to evaluate performance on points of interest $x^*$, which illustrates that 3S closer approximates an oracle both on average and worst case.

| Model | Mean performance difference ↓ | | Worst-case performance difference ↓ | |
|---|---|---|---|---|
| | $\mathcal{D}_{syn}$ (3S) | $\mathcal{D}_{test,f}$ | $\mathcal{D}_{syn}$ (3S) | $\mathcal{D}_{test,f}$ |
| MLP | **0.083** | 0.15 | **0.482** | 0.60 |
| RF | **0.086** | 0.18 | **0.256** | 0.50 |
| GBDT | **0.093** | 0.18 | 0.50 | 0.50 |

**Takeaway.** 3S 's synthetic data can more robustly estimate performance on individual points of interest based on the samples generated in the neighborhood of $x^*$.

### E.2.2 Subgroups as high- and low-density regions

**Motivation** Models often perform worse on outliers and low-density regions due to the scarcity of data during training. We generate insight into this by defining subgroups in terms of density.

**Methodology.**

We would like to partition the points into sets of most likely to least likely w.r.t. density. We use the notion of $\alpha$-support from [90], namely:

$$\text{Supp}^\alpha(p) = \arg\min_{S \subseteq \tilde{\mathcal{X}}} V(S) \ s.t. \ p(\tilde{X} \in S) = \alpha, \tag{3}$$

with $V$ some volume measure (e.g. Lebesgue). In other words, $\alpha$-support $\mathrm{Supp}^\alpha(p)$ denotes the smallest possible space to contain $\tilde{X}$ with probability $\alpha$—which can be interpreted as the $\alpha$ most likely points. Subsequently, we can take a sequence of quantiles, $(q_i)_{i=0}^k \in [0, 1]$, with $q_i = \frac{i}{k}$ and look at the sequence of support sets, $(\mathrm{Supp}^{q_i}(p))_{i=0}^k$.

The $\alpha$-support itself always contains the regions with the highest density. To actually partition the points into sets from likely to unlikely, we instead look at the difference sets. That is, let us define sets $S_i = \mathrm{Supp}^{q_i}(p) \backslash \mathrm{Supp}^{q_{i-1}}(p)$ for $i = 1, ..., k$.

In fact, we do not know $p$ exactly and even if we did, it is usually intractable to find an exact expression for the $\alpha-$support. Instead, we compute the $\alpha$-support in the generative model's latent space, and not the original space. The latent distribution is usually chosen as a $d_z$-dimensional standard Gaussian, which has two advantages: (i) the distribution is continuous—cf. the original space, in which there may exist a lower-dimensional manifold on which all data falls; and (ii) the $\alpha$-support is a simple sphere, with the radius given by $\mathrm{CDF}^{-1}_{\chi^2_{d_z}}(\alpha)$.

**Set-up.** We train a generative model $G : \mathcal{Z} \to \tilde{\mathcal{X}}$ as before, where the input $\mathbf{z}$ is $d_z$-dimensional Gaussian noise. During generation we save inputs $\{z_j\}_{j=1}^{kN}$ and generate corresponding data $\mathcal{D}_{syn} = \{\tilde{x}_j\}_{j=1}^{kN}$. Sets for likely and unlikely points are defined using $\alpha$-support in latent space $\mathcal{Z}$, for quantiles $\mathbf{q} = (q_i)_{i=1}^k$—see Section E.2.2. Specifically, let the index sets for each quantile set be given by $I_i = \{j \mid \|z_j\|_2 < \mathrm{Quantile}(\{\|z_j\|_2\}, q_i)\}$, giving synthetic sets $\mathcal{D}_{syn}^{q_i} = \{\tilde{x}_j \in \mathcal{D}_{syn} \mid j \in I_i\}$ for all $i$. We then evaluate the predictive model $f$ on each $\mathcal{D}_{syn}^{q_i}$ and plot results w.r.t. $\mathbf{q}$. We use the SEER dataset.

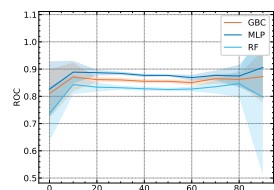

Figure 20: Performance on likely vs unlikely data, where error bars are over 10 runs

**Analysis.** Fig. 20 shows that for the tail quantiles (unlikely) data that the model performance is indeed highly variable and prone to possible poor performance. This is in contrast to the likely data, which has much more stable performance.

**Takeaway.** 3S 's synthetic data can be used to quantify model performance on unlikely and likely data.

### E.3 Improving $G$ when training data of $f$ is available

**Motivation.** In previous experiments, we have assumed that we have access to only $\mathcal{D}_{test,f}$ for training $G$. In some scenarios, we have access to $\mathcal{D}_{train,f}$. For example, the model developer has access to $\mathcal{D}_{train,f}$. Since the training dataset is often larger than the testing dataset, we could use this data to train $G$. Evidently, this results in some bias: we are now using synthetic data generated using $G$, which is trained on $\mathcal{D}_{train,f}$, to evaluate a predictive model that is also trained on $\mathcal{D}_{train,f}$. However, here we show that this bias is outweighed by the improved quality of $G$.

**Setup.** This experiment evaluates the mean performance difference for (i) 3S , (ii) 3S +, and (iii) $\mathcal{D}_{test,f}$. We follow the same setup as the granular subgroup experiment in Section 5.1, with the only difference being that $\mathcal{D}_{train,G} = \mathcal{D}_{train,f}$. We assess the utility of this setup for granular subgroup evaluation.

**Analysis.** Table 15 illustrates the performance similar to that of the main paper. We find that 3S mostly provides a more accurate evaluation of model performance (i.e. with estimates closer to the oracle) compared to a conventional hold-out dataset. This is especially true for the smaller subgroups for which synthetic data is indeed necessary. Furthermore, 3S has a lower worst-case error compared to $\mathcal{D}_{test,f}$. The results with 3S + also illustrate the benefit of augmenting real data with synthetic data, leading to lower performance differences.

Table 15: Mean absolute performance difference between predicted performance and performance evaluated by the oracle, where $G$ is trained on $\mathcal{D}_{train,f}$. 3S better approximates true performance on minority subgroups, compared to test data alone.

| Model | Subgroup (%) | Mean performance diff. ↓ | | | Worst-case performance diff. ↓ | | |
|---|---|---|---|---|---|---|---|
| | | 3S | 3S + | $\mathcal{D}_{test,f}$ | 3S | 3S + | $\mathcal{D}_{test,f}$ |
| RF | #1 (86%) | $6.05 \pm 0.72$ | $2.33 \pm 0.42$ | $9.16 \pm 4.48$ | 7.08 | 2.84 | 11.65 |
| | #2 (9%) | $4.16 \pm 0.33$ | $4.29 \pm 0.47$ | $5.85 \pm 3.18$ | 4.44 | 4.82 | 8.94 |
| | #3 (3%) | $2.15 \pm 1.04$ | $1.88 \pm 0.8$ | $13.59 \pm 5.34$ | 3.41 | 3.03 | 19.73 |
| | #4 (1%) | $4.54 \pm 0.33$ | $4.54 \pm 0.34$ | $7.1 \pm 3.93$ | 4.89 | 4.96 | 13.23 |
| | #5 (1%) | $3.15 \pm 0.77$ | $3.06 \pm 0.77$ | $8.72 \pm 0.0$ | 4.2 | 4.12 | 8.72 |
| GBDT | #1 (86%) | $7.37 \pm 1.09$ | $3.25 \pm 0.32$ | $1.19 \pm 0.81$ | 9.22 | 3.6 | 2.42 |
| | #2 (9%) | $3.44 \pm 0.52$ | $3.37 \pm 0.54$ | $2.56 \pm 1.08$ | 3.98 | 4.02 | 4.49 |
| | #3 (3%) | $2.6 \pm 1.53$ | $2.46 \pm 1.38$ | $4.53 \pm 1.04$ | 4.29 | 3.97 | 5.73 |
| | #4 (1%) | $0.76 \pm 0.58$ | $0.76 \pm 0.57$ | $6.51 \pm 5.07$ | 1.31 | 1.34 | 14.53 |
| | #5 (1%) | $2.23 \pm 0.82$ | $2.15 \pm 0.85$ | $7.73 \pm 3.33$ | 3.47 | 3.45 | 9.4 |
| MLP | #1 (86%) | $6.72 \pm 1.2$ | $3.0 \pm 0.46$ | $1.01 \pm 0.62$ | 8.55 | 3.5 | 1.92 |
| | #2 (9%) | $4.32 \pm 0.46$ | $4.13 \pm 0.54$ | $2.45 \pm 1.46$ | 4.91 | 4.87 | 4.41 |
| | #3 (3%) | $1.98 \pm 1.1$ | $1.88 \pm 0.99$ | $4.06 \pm 0.8$ | 3.58 | 3.33 | 4.69 |
| | #4 (1%) | $2.57 \pm 0.5$ | $2.49 \pm 0.53$ | $7.44 \pm 3.17$ | 3.37 | 3.36 | 12.42 |
| | #5 (1%) | $2.17 \pm 0.62$ | $2.13 \pm 0.64$ | $4.3 \pm 3.66$ | 3.24 | 3.22 | 8.72 |
| SVM | #1 (86%) | $6.74 \pm 0.58$ | $3.0 \pm 0.49$ | $1.26 \pm 0.93$ | 7.37 | 3.6 | 2.46 |
| | #2 (9%) | $5.62 \pm 0.14$ | $5.31 \pm 0.4$ | $3.63 \pm 1.25$ | 5.83 | 5.81 | 5.63 |
| | #3 (3%) | $1.09 \pm 0.73$ | $1.04 \pm 0.73$ | $2.51 \pm 1.09$ | 2.43 | 2.33 | 3.88 |
| | #4 (1%) | $1.08 \pm 0.4$ | $0.97 \pm 0.39$ | $11.62 \pm 4.31$ | 1.51 | 1.44 | 16.01 |
| | #5 (1%) | $4.22 \pm 0.6$ | $4.17 \pm 0.61$ | $3.1 \pm 2.99$ | 5.2 | 5.18 | 6.71 |
| AdaBoost | #1 (86%) | $6.89 \pm 0.96$ | $3.06 \pm 0.31$ | $1.11 \pm 0.71$ | 8.61 | 3.44 | 2.14 |
| | #2 (9%) | $3.91 \pm 0.3$ | $3.76 \pm 0.44$ | $2.31 \pm 1.86$ | 4.24 | 4.31 | 5.15 |
| | #3 (3%) | $2.13 \pm 1.43$ | $2.01 \pm 1.3$ | $4.59 \pm 2.48$ | 4.22 | 4.0 | 7.79 |
| | #4 (1%) | $1.24 \pm 0.65$ | $1.16 \pm 0.69$ | $8.66 \pm 6.46$ | 1.99 | 1.97 | 19.89 |
| | #5 (1%) | $3.42 \pm 0.71$ | $3.36 \pm 0.73$ | $5.17 \pm 3.55$ | 4.47 | 4.44 | 8.05 |
| Bagging | #1 (86%) | $5.81 \pm 1.02$ | $2.65 \pm 0.53$ | $10.01 \pm 4.49$ | 7.03 | 3.61 | 12.83 |
| | #2 (9%) | $4.89 \pm 0.68$ | $5.06 \pm 0.44$ | $7.65 \pm 3.22$ | 5.97 | 5.54 | 10.69 |
| | #3 (3%) | $3.72 \pm 1.65$ | $3.37 \pm 1.64$ | $8.5 \pm 4.77$ | 5.98 | 5.67 | 14.55 |
| | #4 (1%) | $7.01 \pm 0.28$ | $7.0 \pm 0.3$ | $6.16 \pm 3.51$ | 7.46 | 7.48 | 11.21 |
| | #5 (1%) | $5.06 \pm 1.09$ | $5.0 \pm 1.09$ | $4.41 \pm 4.07$ | 6.59 | 6.55 | 9.4 |
| LR | #1 (86%) | $6.51 \pm 0.63$ | $2.97 \pm 0.3$ | $1.09 \pm 0.41$ | 7.57 | 3.38 | 1.81 |
| | #2 (9%) | $4.11 \pm 0.32$ | $3.9 \pm 0.51$ | $3.33 \pm 0.98$ | 4.52 | 4.52 | 4.57 |
| | #3 (3%) | $1.15 \pm 0.73$ | $1.11 \pm 0.64$ | $4.6 \pm 2.72$ | 2.28 | 2.08 | 7.61 |
| | #4 (1%) | $1.69 \pm 0.59$ | $1.62 \pm 0.62$ | $7.16 \pm 4.27$ | 2.57 | 2.55 | 13.48 |
| | #5 (1%) | $4.08 \pm 0.76$ | $4.02 \pm 0.78$ | $4.76 \pm 3.21$ | 5.44 | 5.41 | 7.38 |

# F  Example Model Report

Below we present an example of the type of model report that could be produced when evaluating models using 3S testing.

**Dataset**: Adult [57].

**Intersectional model performance matrix**: diagnosing at a granular level.

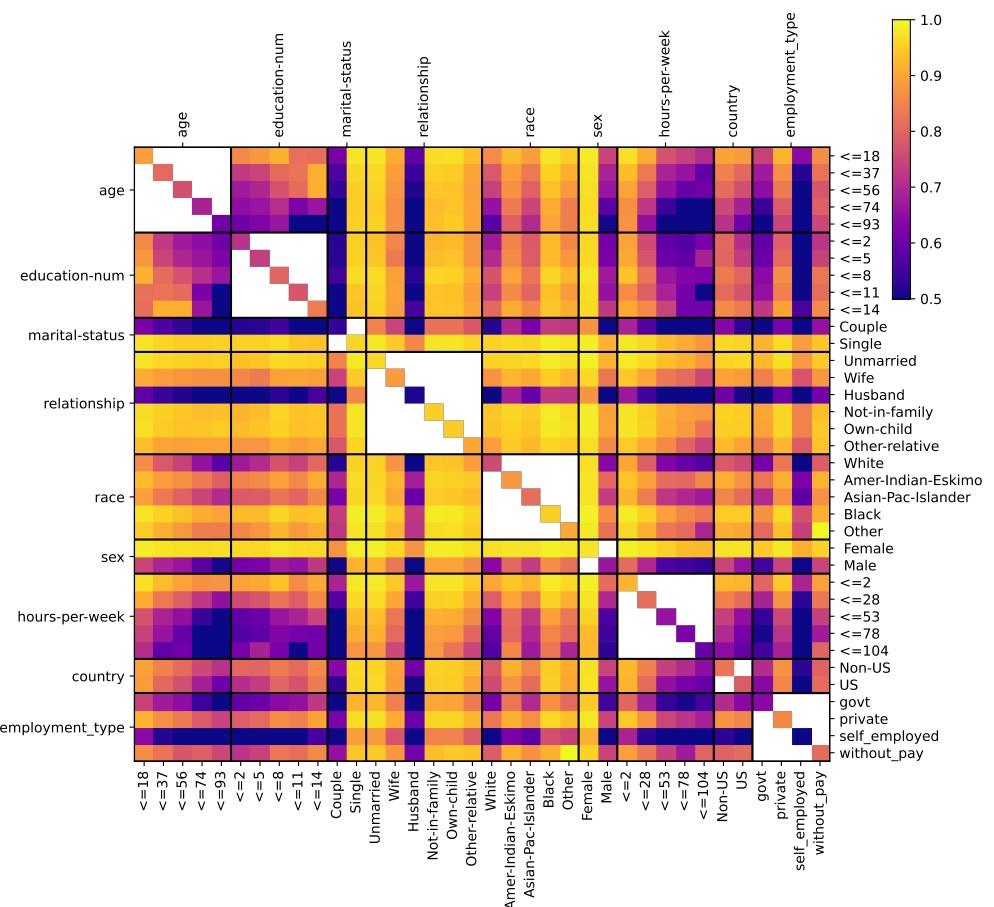

Figure 21: Intersectional performance matrix for the RF model, which diagnoses underperforming 2-feature subgroups (darker implies underperformance).

**INSIGHT**: Model underperformance on self-employed who work more than 80 hours a week. (bottom right arrow)

**Model sensitivity curves: helping to understand performance trends for shifts across the operating range**:

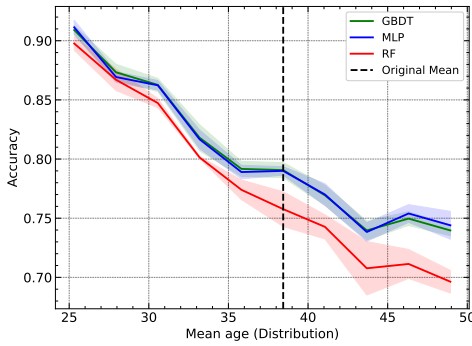

(a) Age: Performance decreases as mean age increases.

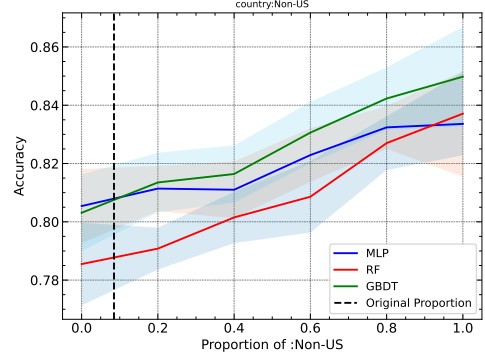

(b) Country of origin: performance increases as the proportion of non-US individuals increases

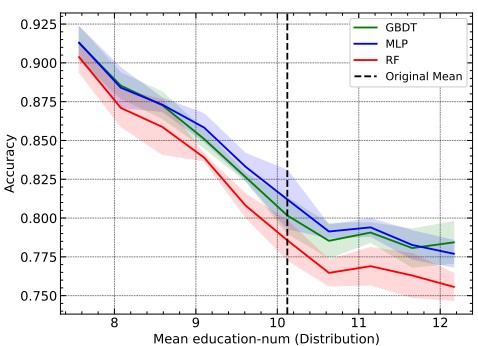

(c) Number of years of education: Performance decreases as mean number of education years increases.

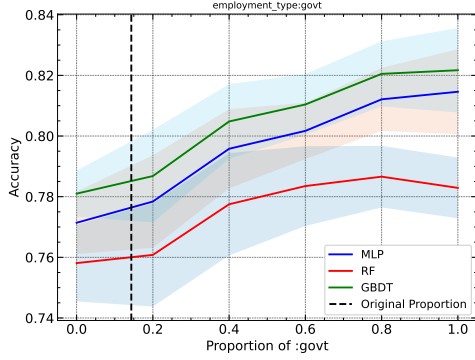

(d) Employment type (government): performance increases as the proportion of government employed individuals increases

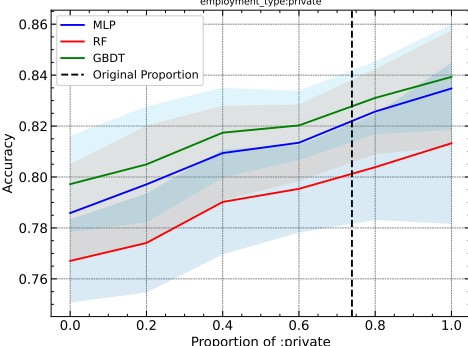

(e) Employment type (private): performance increases as the proportion of private employed individuals increases

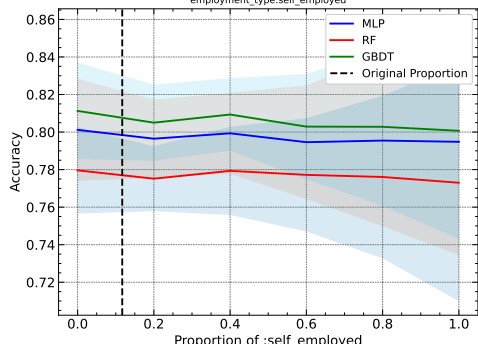

(f) Employment type (self-employed): performance is consistent even as proportion of self-employed individuals increases

Figure 22: Model sensitivity curves for different features, illustrating the relationships/model performance across the operating range.

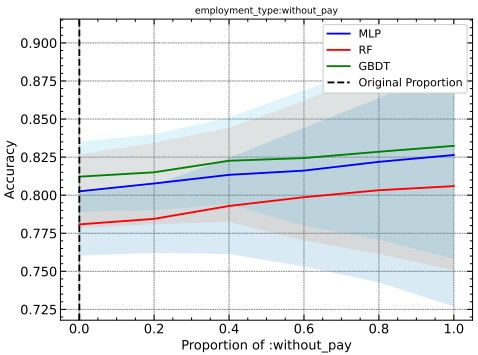

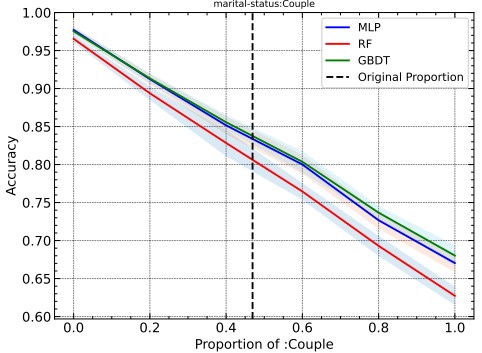

(g) Employment type (without pay): performance is consistent even as proportion of without-pay individuals increases

(h) Marital status (couple): performance decreases as the proportion of married individuals increases

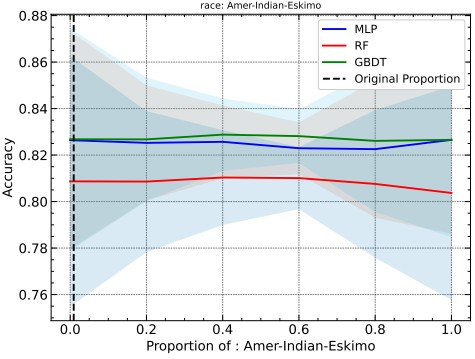

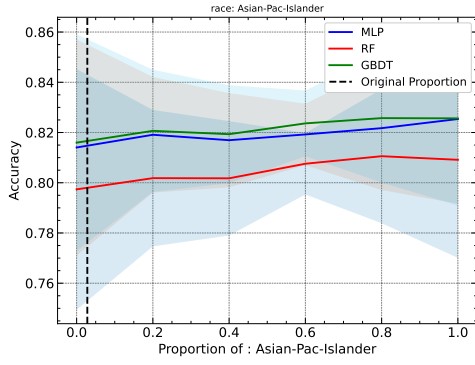

(i) Race (American-Indian-Eskimo): performance remains consistent even as the proportion of American-Indian-Eskimo individuals increases

(j) Race (Asian-Pacific-Islander): performance remains consistent even as the proportion of Asian-Pacific-Islander individuals increases

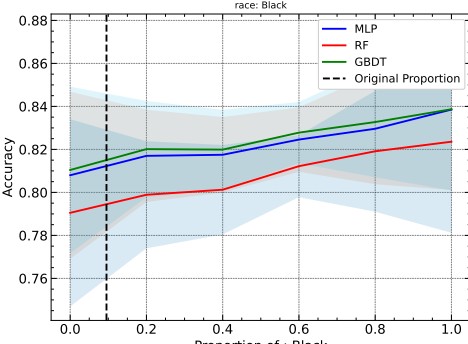

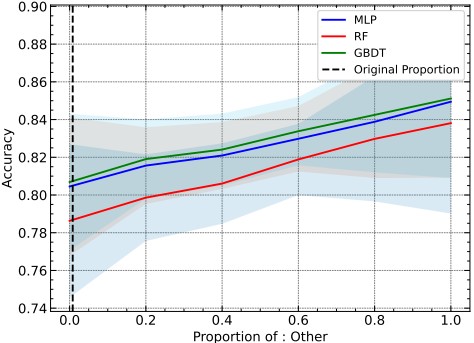

(k) Race (Black): performance increases as the proportion of black individuals increases

(l) Race (Other): performance increases as the proportion of other individuals increases

Figure 22: Model sensitivity curves for different features, illustrating the relationships/model performance across the operating range.

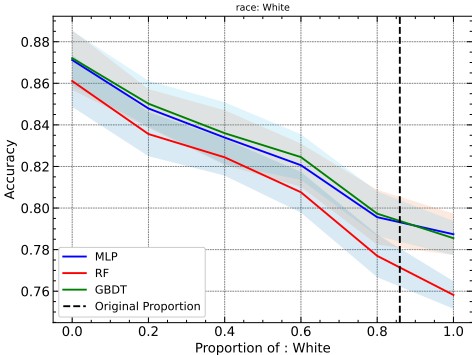

(m) Race (White): performance decreases as the proportion of white individuals increases

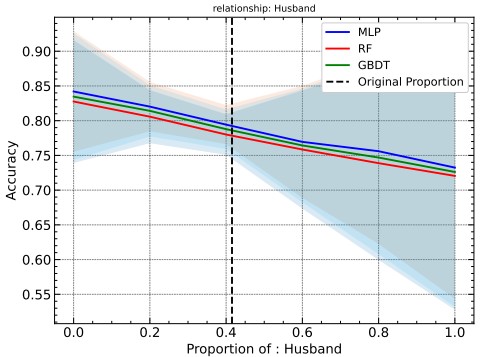

(n) Relationship (Husband): performance decreases as the proportion of individuals classed as husbands increases

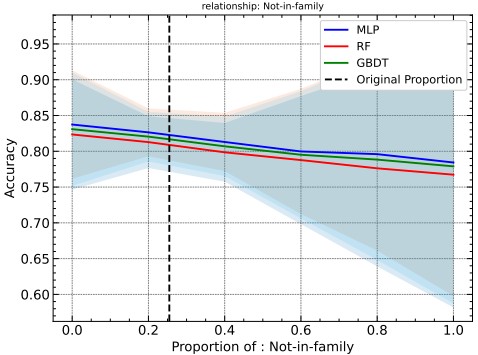

(o) Relationship (Not-in-family): performance decreases as the proportion of individuals classed as Not-in-family increases

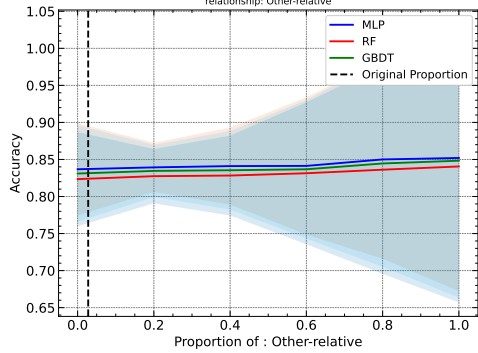

(p) Relationship (other-relative): performance remains consistent as the proportion of individuals classed as other-relative increases

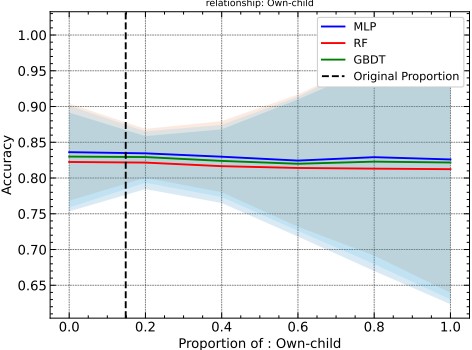

(q) Relationship (Own-child): performance remains consistent as the proportion of individuals classed as Own-child increases

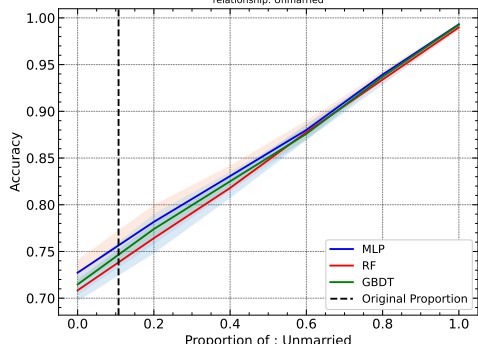

(r) Relationship (Unmarried): performance increases as the proportion of individuals classed as Unmarried increases

Figure 22: Model sensitivity curves for different features, illustrating the relationships/model performance across the operating range.

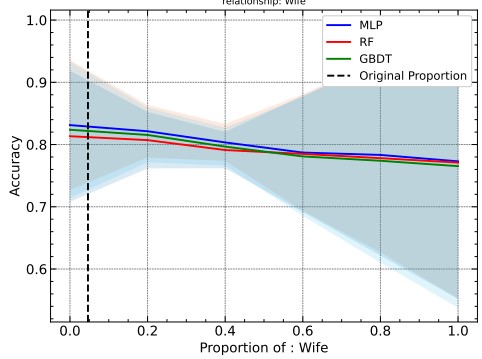
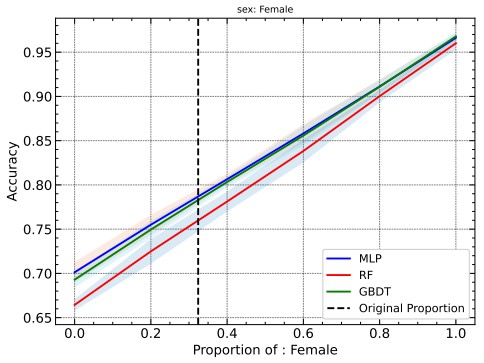

(s) Relationship (Wife): performance decreases as the proportion of individuals classed as Wife increases

(t) Sex (Female): performance increases as the proportion of females increases

Figure 22: Model sensitivity curves for different features, illustrating the relationships/model performance across the operating range.