# OpenReview forum: "Can You Rely on Your Model Evaluation? Improving Model Evaluation with Synthetic Test Data"
_NeurIPS.cc/2023/Conference — NeurIPS 2023 poster_

### Official Review · Reviewer_6cRw · 2023-06-25

**Soundness:** 2 fair
**Presentation:** 2 fair
**Contribution:** 3 good
**Rating:** 5
**Confidence:** 3

**Summary:**

This paper contributes a new model evaluation framework called 3S-Testing, which uses (conditional) deep generative models to create synthetic test sets. To provide uncertainty estimations, 3S uses deep generative ensemble method. It is empirically confirmed that the better performance on small subgroups (with real data), and distributional shifted data.

**Strengths:**

The motivation is clear by targeting the subdominant class. Using generative models (and generative ensembles) to evaluate model is an encouraging attempt, especially for uncertainty estimation. The experiment section is clearly organized and easily understood.

**Weaknesses:**

$\bullet$ The major concern is to evaluate a model, more uncertified models are introduced even though they are used for uncertainty estimation. The author(s) gave reasoning from line 155 to 161. The first is reasonable, while the second is not convincing. For example, in figure 2, the samples of green star class are sparse in the test set by nature. Generating samples of green star class becomes (b) is a “bias” introduced by the generative model.

$\bullet$ 4.2 is hard to read. While the intention was to keep the discussion for shift as general, more and more notations and assumptions for simplifications are continuously added, sometimes without explanation (see below in questions).


**Questions:**

$\bullet$ How to estimate the biases of the generative models on the subgroup, besides providing uncertainty estimation? One possible case is mentioned in weakness, where the generative model generates new samples based on some its own bias.

$\bullet$ Lines 195-198 is confusing to me. After G is trained, how exactly is it used? Steps (2) to (4) are not clear about this. I understand it is talked about later, but it is important to make it clear in the summary paragraph.

$\bullet$ Lines 207-208, Where does that assumption come from?

$\bullet$ Can the author(s) please briefly summarize the discussion (with minimum information/notation/assumptions needed) for lines 204-218? More specifically, does the final recipe proposed in lines 217 to 218 rely on the assumption “simplest such shift” (line 212)?


**Limitations:**

Fully discussed with explanation.

---

> ### Author Rebuttal · Authors · 2023-08-09
>
> Dear reviewer,
>
> Thank you for your time and constructive feedback. We wish to clarify each point in turn.
>
> ## 1. Reasons for using generative models.
> The paper only uses a deep generative ensemble (DGE) [24] to model the uncertainty over the generative model parameters and estimate generative errors. We acknowledge that estimating errors is hard, and this ensemble approach does not come with guarantees—to do so we would instead need to model the intractable posterior over the generative model parameters. Nonetheless, though the 3S estimate may or may not be unbiased, we believe our consistently better evaluation results give strong evidence it can provide a significantly better bias-variance trade-off than real data evaluation, and in turn we hope to have shown the value of synthetic data for evaluation purposes.
>
> **Figure 2b**. Putting this in the context of Figure 2b: we agree that just because the manifold seems plausible, this does not mean it is the true underlying distribution. A different member of the DGE would have converged to a slightly different manifold, caused by a lack of evidence for preferring one manifold over another. By using DGE, we aim to implicitly model our uncertainty over the “correct” manifold.
>
> **Generative models for implicit interpolation/weighting of real data**. At last, the paper’s second argument (L158) for using generative models for evaluation (for which you raise concerns) extends further than learning manifolds. For example, assume you want to estimate the expected performance of model $f$ for a specific person with features $x_c^*$ (a subset of all features). This is hard with real data alone: for continuous variables $X_c$, the probability that there exist points in $D_{test,f}$ with exactly $x_c^*$ is zero, hence $A(f; D_{test,f}, (x:x_c=x_c^*))$ is undefined. There may be many points in $D_{test,f}$ with $x_c$ close to $x^*_c$, but which metric do we use to define “close”, and given this metric, how do we weigh close points to compute $A$? This is non-trivial, because the target distribution may be very sensitive to some feature changes, while being independent of others (hence an Euclidean metric may not be good). Here too, we may expect that a generative model can actually do better. The generative model can “interpolate” the distribution over the whole space, and hence allows generating points with $x_c = x^*_c$. Of course, this may not be perfect, but again the DGE (where each member interpolates differently) gives insight into the uncertainty.
>
> ## 2. Section 4.2 unclarities
> ### L195-198
> **To ensure readers are not confused, we will include the simplified recipe first and move the general shift to the end of the section.**
>
> **General shift**. We can envision many types of shifts that we do not cover in the experiments, but which are also supported by the recipe on line 195 (e.g. concept drifts, see footnote 1 and [Varshney, 2021]). Regardless, we agree that the paper would be clearer if we are not too general when starting Section 4.2 and instead focus on our main setting of interest—i.e. where we make the assumptions of paragraph “Defining shifts” (specifically L204) and where the trained $G$ approximates $p(X_\bar{c}|X_c)$.
>
> **How is $G$ used in practice**. In this case, the 3S evaluation recipe simplifies to: (i) train $G$ to approximate $p(X_\bar{c}|X_c)$, (2) choose a shifted distribution $p(^\sim X_c)$---e.g. a marginal mean shift of the original $p(X_c)$ (Section 5.2.1), or drawing $x_c$ samples from a secondary dataset (Section 5.2.2); (3) draw samples $x_c$, and subsequently *use $G$ to generate the rest of the variables $X_{\bar{c}}$ conditional on these drawn samples*; and (4) evaluate downstream models.
>
> ### L207-208
> We use this assumption because it captures two of the most studied assumptions in distributional shift literature; $p(X|Y)$ is fixed and $p(Y)$ changes, or $p(X)$ changes and $p(Y|X)$ is fixed (see lines 201-203, and [Varshney, 2021; Section 9.2.1]). We have rewritten footnote 2 to make this link more explicit.
>
> ### L204-218
> In summary, we focus on shifts in which some variables’ distribution changes, but the other variables’ distribution conditional on these variables does not. From L211 we discuss the simplest such shift (where $X_c$ is a single variable), because it leads to easily-graspable insight for users. The final recipe (L217) does not depend on this simplest shift. Let us elaborate.
>
> In paragraph L211 we focus on simple one-dimensional shifts (in a single variable $X_c=X_i$), because it is easier to define, visualise, and understand the effects of these shifts hence get meaningful insight into the model. For example, in 5.2.1 we shift the mean of $X_i$, use $G$ to generate data conditional on the shifted $X_i$, and study the effect on downstream model scores; if we had done this with two shifted variables instead, this would require generating and testing over a 2D set of shifted distributions—possible, but not as easy to understand.
>
> Choosing a shift in a multi-dimensional $X_c$’s distribution is not always hard however. For example, as shown in Section 5.2.2, if we observe some of the same features in another dataset, we can simply draw $X_c$ from this dataset and avoid having to define the shifted distribution explicitly. To do so, we assume the variables we observe ($X_c$) accurately describe the shift between our original domain and the new domain, such that the generated data will indeed resemble the target domain.
>
> In any case, the recipe in L217 is independent of the shift distribution $p(^\sim X_c)$. A reasonable constraint is that the shifted distribution “fails within” the old distribution (i.e. $p(X_c)$ dominates $p(^\sim X_c)$). E.g., if $G$ generates medical data based on age and it has only ever seen patients up to 100 years old, it will yield unexpected behaviour when generating 110-year-old patients.
>
> ## References
> Varshney KR. Trustworthy machine learning. 2021. Chappaqua, NY. p. 118.

---

> > ### Comment · Reviewer_6cRw · 2023-08-11
> > **Thank you for your explanations!**
> >
> > Thank you for your reply. The explanations are much clearer than what was written in the paper.
> >
> > According to the explanation, G is not learning p(X) as stated in line 196, but a conditional probability. It aims to generate more (corrupted maybe) samples depend on the shift, doesn't it?
> >
> > Please consider using these simplified and clearer explanation when editing the draft. I have raised the score. Thank you for your explanation and good luck.

---

> > > ### Author Response · Authors · 2023-08-12
> > >
> > > Dear reviewer,
> > >
> > > We are glad our response clarified matters, and would like to thank you for raising your score.
> > >
> > > Indeed, for the shifts and subgroups considered in the paper, it is more efficient to use a conditional generative model—this allows us to directly generate more samples depending on the shift or subgroup. We could use an unconditional generative model, but this would require generating a lot more data and then using post-hoc sampling to satisfy the subgroup/shift definition.
> > >
> > > We agree that the simplified explanation (from our previous response) would be clearer in the context of the rest of the paper and will use it to start Section 4.2.
> > >
> > > We appreciate your time taken to improve the paper!

---

### Official Review · Reviewer_vjS3 · 2023-07-03

**Soundness:** 3 good
**Presentation:** 3 good
**Contribution:** 3 good
**Rating:** 6
**Confidence:** 4

**Summary:**

The authors propose to use synthetic data to evaluate models, especially under distribution shifts or in areas of the input space with low coverage. The authors use CTGAN to empirically validate their idea, and apply it to tabular data.

The paper is a resubmission from ICLR 2023 ( https://openreview.net/forum?id=J7CTp-jNyJ ).

**Strengths:**

Originality: While the general idea of using synthetic data for evaluation does not appear new, to the best of my knowledge this is the first thorough evaluation of the idea for tabular data. (I went out of my way to find published studies on this, and the only works I found were on relatively low-quality journals, or very different in scope).

Quality: Experiments were done on 6 datasets of relatively small scale. This gives a first idea that the idea might work, but mention two possible extensions under "Weaknesses".

Clarity: the paper clarity was fine.

Significance: I think this is an important topic that deserves more study, the results of which will be of interest primarily to practitioners, but might also encourage further research on the topic.

**Weaknesses:**

* The paper mostly focuses on 6 small datasets. While the approach itself is probably interesting in cases where data is scarce, it would be interesting to see this applied to harder datasets that have more than just a couple of dozen features, or where there are millions of samples involved. I personally remain unconvinced the method scales to larger data, and would appreciate if the authors could report results on larger datasets.

* The authors write "* "an end-user only has access to a single draw of Dtest,f . e.g., we might incorrectly overestimate 265 the true performance of minorities. The use of synthetic data solves this.". I'd like to challenge that statement: I think it's very typical in real life (especially given the relatively small data sets that are the focus in the publication) to at least use cross validation to get multiple test sets, potentially even Leave-1-Out CV. This should give a much better estimate of the loss landscape. I'd appreciate if the authors could add such experiments as baselines.

* It would be nice if the authors gave an indication of run times of the various methods they compare too, as this might be useful to practictioners. E.g. They mention in Appendix section C that the GAN was trained with a fairly large hyperparameter search, which I expect is not cheap.

**Questions:**

Figure 4: It is unclear what the x-axis actually represents. It would be nice to get an explanation of what the Different Groups mean. The way it is presented, I don't understand what I am looking at.

Line 180: "similar in vain to Deep Ensembles"  => similar in vein

**Limitations:**

The authors mention limiations in passing, but have not gone out of their way to find out failure modes of their method.

---

> ### Author Rebuttal · Authors · 2023-08-09
>
> Dear reviewer,
>
> Thank you for your time and feedback. We would like to address each of your comments in turn.
>
> ## 1. Large datasets and other limitations
>
> We agree that the paper would benefit from the inclusion of specific failure cases, including very large datasets for which 3S may not be necessary. We discuss this failure case in the general response under “Limitations and Failure Cases”, as we think this discussion will be of interest to some of the other reviewers too. In a nutshell, we include a new experiment with the already-used Bank dataset (test set 75k+) and see that the benefit of 3S is determined primarily by the size of the subgroup we are trying to measure. If this subgroup is very small, we benefit from 3S; when subgroups get larger, real data evaluation is sufficient. We can use the bootstrap-estimated variance of the real dataset estimate to decide when to use 3S or not.
>
>
>
> ## 2. Cross-validation — why its not applicable to our setting & alternatives
>
> We are in the setting where we have a trained black box predictive model $f$, which could have been trained by someone else. For instance, the increasingly common scenario where a trained model is behind an API, yet we still wish to test the model’s capabilities. An example of this was during the pandemic when prognostic models were built in certain countries and were being tested by external parties who did not have access to the original medical training set for data sharing and patient privacy reasons. In this setting, we can only access the model and not the training dataset (see L112-113) — with the test dataset ($D_{test,f}$) our only available data. Of course, this is a challenging setting for 3S as well, since we train the generative model $G$ on $D_{test}$—as we have no access to a (potentially large) $D_{train}$. Since we do not have access to $D_{train}$, we cannot perform cross-validation to obtain multiple train-test splits.
> Nonetheless, we agree with your suggestion and there are alternative ways to obtain multiple “test sets”. A common approach which would fit our problem setting and help get multiple test sets is **bootstrapping** — which allows us to obtain multiple datasets. We have included a bootstrapping baseline in the form of Model-Based Metrics (MBM) [20] — which does a computationally efficient version of nonparametric bootstrap. We outperform MBM both on performance vs Oracle in Figure 4 and coverage of the true value with intervals in Figure 5. To make this more clear to the reader we will update Sec 5.1.1., clarifying that MBM does bootstrapping and that our rationale for inclusion as a baseline is to be an alternative statistical mechanism for obtaining multiple test sets.
>
>
> ## 3. Run times.
>
> We agree that point made about showing run times, especially since we tune a generative model. Please refer to point 4 in the general response where we outline the computational cost. As a summary,  for typical tabular datasets, the computational requirements are often still very manageable — under 5 min for most datasets and under 40 min for the largest dataset Bank (see Response pdf). Of course, tuning will depend on dataset size and the numbers we report in the table in the response pdf are with respect to the dataset sizes in the main manuscript.
>
> ## 4. Unclear subgroups Figure 4.
>
> We regret that the groups in Figure 4 were unclear. The subgroups correspond to possible values one particular categorical feature can take.. This differs per dataset: for Adult, it is different Race groups; for Drug, Ethnicity groups; for Covid, the Region the patient comes from; for Support; race groups; for Bank, different Employment statuses. Since the specifics of each group are challenging to put in the figure without harming readability, we will add an Appendix that specifies the different groups for each dataset.

---

> > ### Comment · Reviewer_vjS3 · 2023-08-15
> >
> > Thank you for your clarifications, and for adding an additional dataset. After seing all the additional reviews, my thoughts on this manuscript are as follows: using generative models to have more synthetic data is certainly an attractive idea. The area feels under-explored and in need of a very thorough analysis: though I found this idea mentioned very often, I have never seen it very rigorously evaluated. To be candid, this work feels a falls a bit short of my standards for a "very rigorous evaluation": it evaluates on fairly few datasets, and has little exploration of errors in the generative process (e.g. it pretty much glosses over how to pick a suitable generative model). So all in all I feel that my original rating of a "weak accept" is still appropriate: the work is okay and has no major flaws. I think with more effort it could even become a hallmark paper on the topic, but in its current state, it falls a short of that.

---

> > > ### Author Response · Authors · 2023-08-17
> > >
> > > Thank you for your response. We agree that the area of generative data augmentation is underexplored and too often poorly evaluated. We hope to have shown the benefit of synthetic data for testing small subgroups and shifts across a range of tabular datasets (and subgroup definitions). Many thanks again for your time and effort reviewing the paper.

---

### Official Review · Reviewer_zuki · 2023-07-05

**Soundness:** 4 excellent
**Presentation:** 4 excellent
**Contribution:** 3 good
**Rating:** 6
**Confidence:** 4

**Summary:**

This paper proposes to use synthetic test data to improve the estimation of model performance for tabular datasets when insufficient test data is available. Their approach of generating synthetic test data conditioned on subgroups improves performance estimation for underrepresented subgroups and can accurately estimate the model performance under distributional shift towards the underrepresented subgroups.

**Strengths:**

The paper is well-written, and the details are covered by the Appendix.

They have done a good job in showcasing the advantage of their synthetic strategy.


**Weaknesses:**

Their proposed method is limited to the tabular data and data synthesis has been looked at in prior studies, but I think this is a good paper with nice experiments.

**Questions:**

Q1. Extreme underrepresentation of the data in some regions, i.e., small subgroups, could affect the training of the generative model as well. At what point does your method fail, i.e., it is simply as good as Dtest strategy? Can you show this for different datasets studied in Figure4?

Q2. How do you justify the 3S+ performing worse than 3S for the bank dataset especially for decision tree and random forest in Appendix Figure 10 and Figure 4?

---

> ### Author Rebuttal · Authors · 2023-08-09
>
> Dear reviewer,
>
> Thank you for your time and feedback. We would like to address each of your comments in turn.
>
> ## 1. Extreme underrepresentation failure case.
>
> We agree that the paper would benefit from the inclusion of specific failure cases, including very small subgroups. We discuss this failure case in the general response under “Limitations and Failure Cases”, as we think this will be of interest to some of the other reviewers. In particular for small subgroups, we see that a practitioner can use the 3S uncertainty bounds to decide whether to trust an estimate. For extremely small subgroups the uncertainty bounds of 3S are usually larger and may urge a practitioner to gather additional data.
>
> ## 2. Bank dataset 3S and 3S+ performance comparison.
>
> Indeed, in Figure 4, 3S+ (pink) has lower MAE vs the Oracle compared to 3S (green) for the small groups for the random forest classifier. In the Appendix, we see that real data ($D_{test}$) has a very large error when estimating the performance of the decision tree—only implying this particular real test data is ill-suited to probe this particular shallow decision tree. Because 3S+ consists of 3S and real test data, it partly inherits the poor performance from the $D_{test}$ estimate and thus leads to poorer estimates than 3S for this particular instance.

---

> > ### Comment · Reviewer_zuki · 2023-08-12
> >
> > Thanks. I am satisfied with the answers to my questions.

---

> > > ### Author Response · Authors · 2023-08-17
> > >
> > > Thank you! And thanks again for your time and suggestions.

---

### Official Review · Reviewer_fN4G · 2023-07-14

**Soundness:** 4 excellent
**Presentation:** 4 excellent
**Contribution:** 3 good
**Rating:** 7
**Confidence:** 3

**Summary:**

In this paper, the authors propose the utilization of synthetic data for evaluating models and introduce an automated suite of synthetic data generators called 3S. 3S offers two key advantages: it enables reliable and detailed evaluation, and it measures model sensitivity to distributional shifts. The paper explores different scenarios involving 3S, providing valuable insights into the application of synthetic data for model evaluation.

**Strengths:**

1. The paper exhibits a clear and easily comprehensible writing style. The authors conduct a comprehensive examination of relevant literature, effectively summarizing its advantages.
2. The issue investigated in this paper, namely the utilization of synthetic data for evaluating models, is both intriguing and significant.
3. The authors effectively discuss various use cases of 3S, offering insights into the practical utilization of synthetic data for model evaluation.







**Weaknesses:**

1. Further discussion on method limitation is needed.


**Questions:**

No questions.

**Limitations:**

As mentioned, further discussion on method limitation is needed.

---

> ### Author Rebuttal · Authors · 2023-08-09
>
> Dear reviewer,
>
> Many thanks for your time and positive feedback.
>
> We agree that the paper would benefit from an extended limitation section and specific failure cases. We discuss these in the general response under “Limitations and Failure Cases”, as we think this discussion will be of interest to some of the other reviewers too.

---

> > ### Comment · Reviewer_fN4G · 2023-08-10
> >
> > Great. Good luck!

---

> > > ### Author Response · Authors · 2023-08-11
> > >
> > > Thank you! And thanks again for your time and suggestions.
> > >
> > > Regards
> > > Paper 12138 Authors

---

### Official Review · Reviewer_DA3a · 2023-07-26

**Soundness:** 3 good
**Presentation:** 3 good
**Contribution:** 3 good
**Rating:** 6
**Confidence:** 3

**Summary:**

This paper proposes a model evaluation framework, generating the synthetic test to mitigate the challenges of model evaluation with limited real test sets, such as unbalanced subgroups and distribution shifts.

**Strengths:**

The idea of using synthetic data to improve the testing and evaluation of machine learning models is impressive.

This paper analyzes clearly and reasonably the failure of real test data and its corresponding challenge for reliable model evaluation.


**Weaknesses:**

**1. Counterintuition.** Although the empirical benefits of syntenic data are observed in the experimental parts, the inequation of Line 155 is not evaluated theoretically. More precise statements are needed here.

**2. Overfitting of deep generative models.** With limited test data in the small subgroup, deep generative models tend to perform overfitting. In this case, the learned manifold could have a huge gap from the real manifold.

It could be better to show more visualizations in Figure 2. In detail, if we change the test samples in the green subgroup, what will happen when we compare synthetic manifolds and the real manifold?

**3. Failure case.** As a model evaluation framework with synthetic data, which makes a lot of sense in the real world, this paper lacks failure cases to show the limitations of their works.


**Questions:**

Please check the weakness.

**Limitations:**

Please check the weakness.

---

> ### Author Rebuttal · Authors · 2023-08-09
>
> Dear reviewer,
>
> Thank you for your time and feedback. We would like to address each of your comments in turn.
>
> ## 1. Counterintuitive [line 155]
>
> We do not want to give the impression that the equation in line 155 holds in general—though we hope the experiments are convincingly consistent and argue it very often will. To mitigate us seeming to make any theoretical claims here, we have changed line 155 to "It may seem counterintuitive that $|A^*-A(f;D_{syn}, S)|$ **would ever be lower than** $|A^*-A(f;D_{test,f}, S)|$, ...” and we will link to an extended limitation section with failure cases—see point 3 below.
>
> ## 2. Overfitting of deep generative models and learnt synthetic data manifold
>
> We agree that overfitting in generative models is a serious problem, possibly leading to serious errors (e.g. in learnt manifold). In practice however, we have seen that the real manifold seems very well approximated. In the response pdf, Figure 1 we follow your suggestion and include a T-SNE of the real test data, an oracle set, and synthetic data. We observe that the synthetic data displays the same microstructures/manifold as the oracle set.
>
> *DGE*. To mitigate (and provide insight into) generative error, 3S uses a deep generative ensemble (DGE) [24] to model the uncertainty over the generative model parameters. Putting this in the context of Figure 2b: we agree that the learnt manifold may vary in practice, because there is a lack of evidence for preferring one manifold over another—e.g. a different member of the DGE would have converged to a slightly different manifold. By using DGE, we aim to implicitly model our uncertainty over the “correct” manifold.
>
>
> ## 3. Failure cases
>
> We agree that the paper would benefit from an extended limitation section and specific failure cases. We discuss these in the general response under “Limitations and Failure Cases”, as we think this discussion will be of interest to some of the other reviewers too.

---

> > ### Comment · Reviewer_DA3a · 2023-08-16
> >
> > Thank the authors for this detailed response. My concerns are well-addressed. I'd like to improve my score to "weak accept".

---

> > > ### Author Response · Authors · 2023-08-17
> > >
> > > We are happy to hear your concerns have been addressed and will make changes to the revised paper to reflect this discussion. Thank you for reconsidering your score, and thanks again for reviewing our paper.

---

### Author Rebuttal · Authors · 2023-08-09

Dear reviewers,

Some reviewers suggested a longer limitations discussion, others were interested in specific failure cases. We address these points here. In the camera-ready paper we will extend the discussion to reflect these.

## Limitations

We summarise 3S’s main limitations (where applicable referencing limitations already outlined in the paper).

1. **Subgroups in real test set are large**. When the aim is subgroup evaluation and there is sufficient real data for this subgroup, the real data estimate on $\mathcal{D}_{test,f}$ will be sufficiently accurate and there is no need to use 3S given the computational overhead. See *Failure Case 1* for new experiments and further discussion. Note, even for very large datasets, there can be very sparse regions or small subgroups for which using 3S is still beneficial. Also note that this limitation mostly applies to subgroup evaluation and less to Generating Synthetic Data with Shifts (Sec. 4.2), because performance estimates for possible shifts is less trivial using real data alone (e.g. require reweighting or resampling of test data) and we show (Sec. 5.2, Table 1 & Fig. 6) that 3S beats real data baselines consistently.

2. **Possible errors in the generative process** (L177, L402 & Fig. 7). Errors in the generative process can affect downstream evaluation. This is especially relevant for groups or regions with few samples, as it is more likely a generative model does not fit the distribution perfectly here. By replacing the generative model by a deep generative ensemble [24], 3S provides insight into its own generative uncertainty. When there is very little data and 3S’s uncertainty bounds are too large, practitioners should consider gathering additional data. See *Failure Case 2* below.

3. **Not enough knowledge or data to generate realistic shifts** (L398, Sec. 5.2.2). The success of modelling data with distributional shifts relies on the validity of the distributional shift’s assumptions. This is true for 3S as much as for supervised works on distributional shifts (e.g. see [Varshney, 2021; Sec. 9.2.1]). In Sec. 5.2.2 we show how a lack of shift knowledge may affect evaluation. We aimed to generate data to test a model on UK patients, using mostly data from US patients. We do not define the shift explicitly—we only assume that we observe a small number (1 to 4) of features $X_c$ (hence the conditional distribution of the rest of the features conditional on these features is the same across countries). In Fig. 6b, we show the assumption does not hold when we only see one feature—this lack of shift knowledge is a failure case. When we add more features, the assumption is more likely to approximately hold and 3S converges to a better estimate. We reiterate that invalid shift assumptions are a limitation of any distributional shift method, e.g. the rejection sampling (RS) baseline using only real data is worse than 3S overall.

4. **Computational cost** (L404). The computational cost & complexity of using generative models is always higher than using real test data directly. For typical tabular datasets, the computational requirements are often very manageable: under 5 min for most datasets and under 40 min for the largest dataset Bank (see Response pdf). The reported times correspond to the dataset sizes in the paper.  We would like to emphasise that the cost at deployment time for a poorly evaluated model can be unexpectedly high, which warrants 3S’s higher evaluation cost.
Additionally, pre-implemented generative libraries (e.g. Patki 2016, Qian 2023) can accelerate the generative process, automating generator training with minimal user input.

## Failure cases

For limitations 1 & 2 mentioned above, we include two new experiments highlighting failure cases on two extreme settings:

### F1. Subgroups in real test set are large
With sufficiently large real data, 3S provides no improvement despite higher complexity. We can determine sufficient size by estimating the variance Var$(A)$ of the performance metric $A(f;D,S)$ w.r.t. _the random variable_ denoting the test data $D$. With only access to one real dataset $D_{test}$ however, we can approximate $Var(A)$ via bootstrapping (Fig. 5). If Var$(A)$ falls below a small threshold $\alpha$ (e.g. 0.05), practitioners may for example decide to trust their real data estimate and not use 3S, as further evaluation improvements are unlikely. In our Bank dataset experiment (500k examples), we vary age & credit risk thresholds, retaining samples above each cut-off to shrink the test set (**see Fig. 2 - response pdf**). Note that for large datasets but very small subgroups, the $D_{test,f}$ estimate still has a high variance, (reflected in the large bootstrapped Var$(A)$ ), hence this should urge a practitioner to use 3S (see Fig. 2 response pdf).

### F2. Large uncertainty for very small test sets
At the other extreme, when there are _too few_ samples the uncertainty of 3S (quantified through a DGE [24]) can become too large. We include a new experiment (**Fig. 3, response pdf**) in which we reduce subgroups to fewer than 10 samples in the test data. We train $G$ in 3S on the overall test set (which includes the small subgroup) with $n_{samples}$. Despite good performance versus an oracle, the uncertainty intervals from 3S's ensemble span 0.1-0.2.  These wide intervals make the 3S estimates unreliable and less useful, and would urge a practitioner to consider gathering additional data. The key takeaway: With extremely sparse subgroups, the large uncertainties signal that more data should be gathered before relying on 3S's uncertain estimates.

## References
Patki, N., Wedge, R., & Veeramachaneni, K. (2016, October). The Synthetic Data Vault. IEEE.

Varshney KR. Trustworthy machine learning. 2021. p. 118.

Qian, Z., Cebere, B. C., & van der Schaar, M. (2023). Synthcity: facilitating innovative use cases of synthetic data in different data modalities. arXiv preprint arXiv:2301.07573.

---

### Decision · Program_Chairs · 2023-09-21

**Decision:**

Accept (poster)

**Comment:**

The paper addresses the evaluation of generative models by the use of synthetic data. Specifically, it focuses on the problem of underrepresented groups and domain shift which are important topics. I believe these topics are important and having a proper evaluation of small subpopulations is important for fairness. The work also addresses some of the limitations of using a generative model by taking care to estimate the generative uncertainty. I believe this work is of value to the research community and should be accepted to this conference.